# Filtered not Mixed: Filtering-Based Online Gating for Mixture of Large Language Models

**Raeid Saqur**[2,4] **Anastasis Kratsios**[3,4] **Florian Krach**[6] **Yannick Limmer**[1] **Jacob-Junqi Tian**[4] **John Willes**[4] **Blanka Horvath**[1] **Frank Rudzicz**[4,5]

[1]Oxford-Man Institute for Quantitative Finance, Department of Mathematics, University of Oxford
[2]Department of Computer Science, University of Toronto
[3]Department of Mathematics, McMaster University
[4]Vector Institute
[5]Faculty of Computer Science, Dalhousie University
[6]Department of Mathematics, ETH Zürich
raeidsaqur@cs.toronto.edu, kratsioa@mcmaster.ca, florian.krach@math.ethz.ch,
{limmery,horvath}@maths.ox.ac.uk, frank@dal.ca

## Abstract

We propose **MoE-F** – a formalized mechanism for combining $N$ pre-trained expert Large Language Models (LLMs) in online time-series prediction tasks. MoE-F adaptively forecasts the optimal weighting of LLM predictions at each time step by leveraging the conditional information in each expert's running performance, enabling the best combination of experts for the next step prediction. Diverging from static (learned) Mixture of Experts (MoE) methods, our approach employs time-adaptive stochastic filtering techniques to combine experts. By framing the expert selection problem as a finite state-space, continuous-time Hidden Markov model (HMM), we can leverage the *Wonham-Shiryaev* filter. Our approach first constructs $N$ parallel filters corresponding to each $N$ individual LLMs. Each filter proposes its best combination of LLMs, given the information that they have access to. Subsequently, the $N$ filter outputs are optimally aggregated to maximize their robust predictive power, and this update is computed efficiently via a closed-form expression, thus generating our ensemble predictor. Our contributions are: **(I)** the MoE-F algorithm – deployable as a plug-and-play filtering harness over any heterogenous mixture of LLMs or specialized models, **(II)** theoretical optimality guarantees of the proposed filtering-based gating algorithm (via optimality guarantees for its parallel Bayesian filtering and its robust aggregation steps), and **(III)** empirical evaluation and ablative results using state of the art foundational and MoE LLMs on a real-world *Financial Market Movement* task based on streaming news where MoE-F attains a *17% absolute and 48.5% relative* F1-score improvement over the best performing individual LLM expert. Further, we provide empirical evidence of substantial performance gains with MoE-F over specialized models in the long-horizon time-series forecasting domain using electricity-grid datasets. Supplementary materials available at: https://github.com/raeidsaqur/moe-f.

## 1 Introduction

Mixture of expert models (MoEs), such as Liu et al. [2024a;b], Guo et al. [2024] the seminal Switch Transformers [Fedus et al., 2022], and more recently Mixtral [Jiang et al., 2024], Gemini [Google], DBRX [MosaicAI, 2024] and many others (e.g. [Saad et al., 2023, Chowdhury et al., 2023, Li et al., 2024, Puigcerver et al., 2024]), have taken a center stage in the generative AI zeitgeist since they allow the number of parameters in large language models (LLMs) to be scaled-up while maintaining a roughly constant computational cost. This is due to the sparse activation strategy employed by MoEs, wherein several offline experts are banked, and a gating network routes each input to a small subset of expert models when generating a prediction [Zhou et al., 2022]. Thus, only a few experts must be loaded into active memory at any given time, to generate a prediction from

any novel input. Currently, most MoE pipelines are designed for static tasks, i.e. tasks without temporal structure; consequentially these pipelines are limited to gating mechanisms which are *constant-in-time*. However, many prediction problems have a temporal structure which is not being leveraged by these classical *constant-in-time* pipelines; examples include time-series data appearing in physics [Mao, 2007], finance [Sharp, 1990], decision science [Feinberg & Shwartz, 2012], or most reservoir computing applications [Jaeger, 2002].

Dynamic prediction problems are fundamentally different from static ones, since each time a new instance arrives a prediction is generated by each expert, hence, the user progressively gains more information as to which expert models are the strongest predictors for a given task. This information (observed measurements) can then be fed back to the MoE's pipeline and used to update its gating mechanisms to select the optimal combination of experts. The key challenge is that we cannot directly observe which is the true best expert(s) to use in predicting the target.

We consider the finite state-space process, which selects the optimal expert, as the unobservable signal process. The problem of best estimating this signal process, given the information from observable measurements, is precisely the continuous-time and finite state-space *stochastic filtering problem* initially studied by Wonham [1964]. The solution to a generalized version of that component of our problem, considered in Liptser & Shiriaev [1977], is a so-called *stochastic (optimal) filter*. It is a closed-form recursion updating the best estimate of the unobservable signal process (the best expert to use in our case) given the observable measurements (the performance of each expert), formalized by a precision for the conditional distribution of the signal process.

Our *Mixture-of-Experts Filter (MoE-F) mechanism* incorporates $N$ stochastic filters, implemented in parallel, to update the gating mechanism, which routes any new input of the mixture model to an optimal combination of our $N$ expert models *given their measured historical performance*. These $N$ predictions generated by the parallel stochastic filters are then robustly aggregated into a single robust mixture prediction by a mechanism similar to those used in PAC-Bayes theory, e.g. Alquier [2008], Rothfuss et al. [2021]. In this way, our proposed MoE-F algorithm optimally predicts the best mixture of expert models while dynamically *adapting* itself to the observed performance of each expert model. We emphasize that this type of dynamic-updating procedure is only possible due to the temporal structure of the time-series prediction tasks we are interested in.

**Contributions.** We construct an online gating mechanism (Algorithm 1) for an online mixture of expert (LLM) models for time-series prediction tasks. Using tools from stochastic calculus and stochastic filtering theory, we prove the optimality of our online gating mechanism's first parallel Bayesian estimation stage (Theorem 1). Using tools from the theory of Markov chains and from Gibbs-measures we demonstrate the optimality of the robust aggregation phase (Theorem 2).

**Outline.** Section §2 covers the necessary background, including our probabilistic framework and continuous-time Markov chains theory. Section §3 explicates the MoE-F algorithm. Section §4 provides theoretical guarantees for the two phases of MoE-F algorithm, with proofs in the appendix. Section §5 demonstrates the large-scale application of our algorithm on real-world online classification and regression tasks. For classification, we evaluate the online prediction of financial market movements using the NIFTY dataset [Saqur et al., 2024] and concurrent SOTA LLM experts. For regression, we showcase the filter's utility in long-term time-series forecasting (LTSF) using the ETTh1, ETTh2 [Zhou et al., 2021] datasets with concurrent specialized LTSF models.

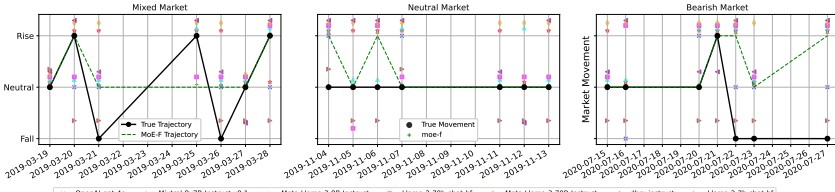

*Figure 1:* A visualization of MoE-F 's application as a filtering harness. Depicts seven SOTA LLMs predicting market movement direction over three randomly sampled windows of seven (trading) days across varying market regimes — **left**: *mixed* market with high fluctuations, **middle**: *neutral*, and **right**: *bearish* market. In all sub-plots, the ground-truth (market) trajectory is in black, and the filtered trajectory is depicted in dotted green. All other experts' (Table 2) predictions are overlaid as scatter-plot points. No values for non-trading days.

## 2 PRELIMINARIES

**Notations for Markov Chains Matrices.** For each $N \in \mathbb{N}_+$, we use $\mathcal{P}_N$ to denote the set of $N \times N$ row-stochastic matrices; i.e. $P \in \mathcal{P}_P$ if each row of $P$ is a vector in $[0, 1]^N$ whose entries sum to 1. We use $\mathcal{P}_N^U$ to denote the set of "uniform row-stochastic matrices", meaning the set of $P \in \mathcal{P}_N$ with identical rows. For each such $N$, we use $\mathcal{Q}_N$ to denote the set of $N \times N$ Markov Q/intensity matrices; that is, $Q \in \mathcal{Q}_N$ if $Q_{n,m} \geq 0$ whenever $n \neq m$, for $n, m = 1, \ldots, N$ and the rows of $Q$ sum to 0. For any such $Q$, the matrix exponential $\exp(tQ)$ is a row-stochastic matrix for every $t \geq 0$. The $\ell^\infty$ norm of an $N \times M$ matrix $X$ will be denoted by $\|X\|_\infty \stackrel{\text{def.}}{=} \max\limits_{\substack{n=1,\ldots,N \\ m=1,\ldots,M}} |X_{n,m}|$.

**Probability Theory.** Let $x_\cdot : [0, \infty) \to \mathbb{R}^d$ be a continuously differentiable path, and consider a complete filtered probability space $(\Omega, \mathcal{F}, \mathbb{F} \stackrel{\text{def.}}{=} (\mathcal{F}_t)_{t \geq 0}, \mathbb{P})$ with right-continuous filtration $\mathbb{F}$ and supporting a 1-dimensional Brownian motion $W_\cdot \stackrel{\text{def.}}{=} (W_t)_{t \geq 0}$. We want to predict a 1-dimensional *target (stochastic) process* $Y_\cdot \stackrel{\text{def.}}{=} (Y_t)_{t \geq 0}$ using our MoE models $F$, as described in § 1. Assume $Y_\cdot$ evolves according to the dynamics in Equation equation (1). The process $w_\cdot$ is assumed to be a hidden Markov process, taking values in the standard basis $\{e_n\}_{n=1}^N$ of $\mathbb{R}^N$. Its evolution is governed by the intensity (or Q) matrix $Q_\cdot : [0, \infty) \to \mathbb{R}^{N \times N}$, which describes the *rate at which the transition probabilities* of the Markov chain $w_\cdot$ change.

Formally, for $i, j = 1, \ldots, N$ and $t, \Delta > 0$ its $(i, j)^{th}$ entry $Q_t^{i,j} \stackrel{\text{def.}}{=} (Q_t)_{i,j}$ is determined by

$$\left| \mathbb{P}(w_{t+\Delta} = e_i | w_t = e_j) - I_{i=j} - \Delta \cdot Q_t^{i,j} \right| \in o(\Delta).$$

We provide additional details on stochastic filtering and matrix operations in Appendix D, that are not required to formulate our main guarantee (Theorem 1) but are included for completeness of proofs.

## 3 THE MoE FILTERING (MoE-F) ALGORITHM

Our setting can be formalized as follows. We consider an input signal $x_\cdot$, a $d$-dimensional smooth path, and the target process $Y_\cdot$, a continuous-time 1-dimensional stochastic process. We are given $N$ auto-regressive, causal pre-trained expert models, $f^{(1)}, \ldots, f^{(N)} : \bigcup_{t \geq 0} C([0, t), \mathbb{R}^d) \to \mathbb{R}$ which map input paths such as $x_\cdot$ to one-dimensional predictions in time such that their predictions do not depend on future states of $x_\cdot$ or $Y_\cdot$ when predicting at any time $t \geq 0$.

Instead of treating the target as being static we allow it to evolve dynamically in time. We assume that there is a *Hidden Markov process* $w_\cdot$ dictating which expert best approximates $Y_\cdot$, up to some Brownian measurement noise $W_\cdot$; thus, we postulate that $Y_\cdot$ evolves according to the stochastic differential equation (SDE) with stochastic drift given by the equation (1):

$$Y_t = Y_0 + \underbrace{\int_0^t w_s^\top F(x_{[0,s)}) \, ds}_{\text{Best Expert Estimate}} + \underbrace{\int_0^t dW_s}_{\text{Idiosyncratic Residual Noise}}, \tag{1}$$

where the Markov process $w_\cdot$ randomly masks *all but one* expert at any given time and concatenates the experts $F(x_{[0,t)}) \stackrel{\text{def.}}{=} (f^{(n)}(x_{[0,t)}))_{n=1}^N$. The first term in equation (1), $\int_0^t w_s^\top F(x_{[0,s)}) \, ds$, represents the true best sparse approximation of $Y_t$ — i.e. the maximally sparse "single expert" — by the ensemble of experts $F$. The $\int_0^t dW_s$ term, is an additive Gaussian noise with mean 0 and variance $t$.

Thus, the mixture coefficients $w_\cdot$ act as an *unobservable signal* which is indirectly observed through the *measurements* recorded by each expert's running performance $\ell^{(n)} \stackrel{\text{def.}}{=} (\ell_t^{(n)})_{t \geq 0}$, where $\ell_t^{(n)}$ is the loss incurred by the $n^{th}$ expert at time $t \geq 0$ as quantified by the chosen loss-function $\ell : \mathbb{R} \times \mathbb{R} \to \mathbb{R}$.

This *online mixture of experts problem* is a two-fold problem which can be solved by our proposed MoE-F procedure. In the first phase, we solve $N$ stochastic filtering problems in parallel, which optimally estimate $w_\cdot$, at any given time $t \geq 0$, given the measurement performance of any one expert $(\ell_s^{(n)})_{0 \leq s \leq t}$. The second phase of the MoE-F algorithm aggregates these optimal predictions.

### 3.1 THE MoE FILTERING

Our MoE-F procedure (Algorithm 1) summarized in Figure 2 operates in two steps, both of which are efficiently computable due to closed-form expressions.

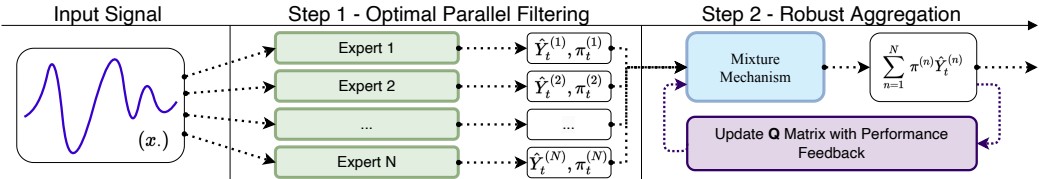

*Figure 2:* **MoE-F** **Mechanism**: conceptual depiction of an input signal $x.$ evolving in $\mathbb{R}^d$ with $N$ experts ($\pi$).

In the first phase of our algorithm, we solve $N$ distinct stochastic filtering problems in parallel. Each $n^{th}$ expert provides an optimal Bayesian prediction $\hat{Y}_t^{(n)}$ of the $Y_t$ in equation (1) based on their running performance $\ell_t^{(n)}$ up to the current time $t \geq 0$. Additionally, each expert generates a *ranking* $\pi_t^{(n)}$ reflecting the reliability of all experts' performance, including its own, up to that point. The optimality of this update is guaranteed by Theorem 1.

Next, these optimal Bayesian predictions from each expert are robustly aggregated by a central module. This module first computes a robust version $\bar{\pi}$ by aggregating the reliability scores of each expert's filter $\{\pi_t^{(n)}\}_{n=1}^N$ and then uses it to compute a robust ensemble prediction $\sum_{n=1}^N \bar{\pi}_t^n \hat{Y}_t^{(n)}$ for the target $Y_t$. Finally, the dynamics of the hidden Markov chain $w.$ in equation (1), encoded by its so-called $Q$/intensity matrix, are re-estimated in a robust fashion using the running performance of each of the $N$ experts up until the current time $t \geq 0$. The optimality of this robust aggregation step is provided by Theorem 2.

**Loss functions.** In this paper, we consider either the binary cross entropy (BCE) or mean-squared error (MSE) as loss functions. These are respectively defined by

$$\ell(\hat{y}, y) \stackrel{\text{def.}}{=} \begin{cases} y \log(\hat{y}) + (1 - y) \log(1 - \hat{y}), & \text{BCE}, \\ |y - \hat{y}|^2, & \text{MSE}, \end{cases}$$

where $y, \hat{y}$ belong to $\mathbb{R}$ in the regression case and to $[0, 1]$ in the classification case. Though the setting in equation (1) directly applies to unbounded targets, like in regression, one can also use it in classification by simply taking $Y.$ to be the logits (inverse logistic transform) of the classification probabilities. Since the logistic function is a bi-measurable bijection, the filtration generated by the logits or the classification probability processes are equal; thus, the filter does not change.

**Helper Functions.** For a concise presentation of our MoE-F algorithm (Algorithm 1), we now introduce a few helper functions: $A$, $\bar{A}$, $B$, $F$. Intuitively, the first two helper functions, $A_t$ and $\bar{A}_t$ compute the gradient and average the gradient of the $n^{th}$ expert's ($\pi^{(n)}$) prediction of the target path $y.$ up to time $t$, accounting for the sensitivity of the $n^{th}$ expert to changes in the input path at time $t$. Here, the averaging is taken uniformly over which "latent expert" is active. We rely on the time-derivative of the loss function evaluated at the $n^{th}$ expert's model, which is given in closed-form

$$A_t^{(n)}(w, y_t) \stackrel{\text{def.}}{=} \begin{cases} -\frac{\left(y_t - f^{(n)}(x_{[0,t)})\right) \Delta f^{(n)}(x_{[0,t)})}{\left(1 - f^{(n)}(x_{[0,t)})\right) f^{(n)}(x_{[0,t)})} - \log\left(\frac{f^{(n)}(x_{[0,t)})}{1 - f^{(n)}(x_{[0,t)})}\right) \left[w_t^\top F(x_{[0,t)})\right], & \text{if } \ell \text{ is BCE}, \\ 2\left(y_t - f^{(n)}(x_{[0,t)})\right)\left(w_t^\top F(x_{[0,t)}) - \Delta f^{(n)}(x_{[0,t)}) + 1\right), & \text{if } \ell \text{ is MSE}. \end{cases} \quad (2)$$

We will also use the concatenated and averaged gradients of the loss of each expert, denoted respectively by $A_t$ and $\bar{A}_t$, and are defined as:

$$A_t(w, Y_t) \stackrel{\text{def.}}{=} \left(A_t^{(n)}(w, Y_t)\right)_{1 \leq n \leq N} \quad \text{and} \quad \bar{A}_t^{(n)}(\pi^{(n)}, y_t) \stackrel{\text{def.}}{=} \sum_{i=1}^N A_t^{(n)}(e_i, y_t) \pi^{(n:i)}, \quad (3)$$

where the sensitivity of the $n^{th}$ expert at time $t$ to changes in the input path $x.$ is

$$\Delta f^{(n)}(x_{[0,t)}) \stackrel{\text{def.}}{=} \lim_{\varepsilon \downarrow 0} \frac{1}{\epsilon}(f^{(n)}(x_{[0,t)}) - f^{(n)}(x_{[0,t-\epsilon)})) \approx f^{(n)}(x_{[0,t)}) - f^{(n)}(x_{[0,t-1)}).$$

Helper function $B$ quantifies the gradient of the loss function (BCE or MSE) of the $n^{th}$ expert's prediction w.r.t. the observed target $y.$, ignoring changes in the input path $x.$, and is given by:

$$B_t^{(n)}(y_t) \overset{\text{def.}}{=} \begin{cases} -\log\left(\frac{f^{(n)}(x_{[0,t)})}{1-f^{(n)}(x_{[0,t)})}\right), & \text{if } \ell \text{ is BCE,} \\ 2^{3/2}\left(y_t - f^{(n)}(x_{[0,t)})\right), & \text{if } \ell \text{ is MSE.} \end{cases} \tag{4}$$

The helper function $F$ concatenates the $N$ experts $F(x_{[0,t)}) \overset{\text{def.}}{=} \left(f^{(1)}(x_{[0,t)}), \ldots, f^{(N)}(x_{[0,t)})\right)$.

As illustrated by Figure 2, our MoE-F (Algorithm 1) operates in two steps. The first step (lines 3-11) implements $N$ parallel stochastic filters yielding optimal Bayesian predictions of $Y.$ given the information available in the running loss function of each of the $N$ expert models. The second step (lines $12-16$) aggregates the $N$ predictions of each expert model in a robust manner, and it performs a bi-level optimization to compute our most robust Markov $Q$/intensity matrix supporting the running performance of each expert thus far.

**Step 1 – Optimal Parallel Filtering.** Lines 3-8 of our MoE-F algorithm is a time discretization of the stochastic optimal filtering equations in equation (5) according to the standard Euler-Maruyama scheme, see e.g. Hutzenthaler & Jentzen [2015]. This is performable in parallel, with each parallel computing branch corresponding to a single expert. As in discrete-time filtering implementations, e.g. Grewal & Andrews [2014], Kim et al. [2018], the discretized infinitesimal change in the running loss of the $n^{th}$ expert is given by $d\ell_{[0,t)}^{(n)} \approx \ell_t^{(n)} - \ell_{t-\Delta}^{(n)}$; provided that $t > \Delta > 0$. Thus, the *innovations process* $\bar{W}$ is the change in $\ell^{(n)}$ over the time increment $[t - \Delta : t]$ minus $\bar{A}(Y_{[0,t)})$, re-normalized by $B_i(Y_{[0,t)})$. Importantly, this means that it is *computable online from the incoming target process* $Y.$. In line 8, each expert (still in parallel) proposes their best estimate $\hat{Y}_t^{(n)}$ for $Y_t$ using their mixture weights, estimated only using their (local) information in their historical loss.

Finally, in line 10, the *reliability* of each expert is measured (still in parallel) by evaluating the loss between the target process $Y_t$ and their best (individual) prediction $\hat{Y}_t^{(n)}$ of it. This reliability metric is summarized by their score $s_n$ of the $n^{th}$ expert.

**Step 2 – Robust Aggregation.**
The prediction quality of each expert $\hat{Y}_t^{(n)}$ has been compressed into a score $s_n$. These scores are used to aggregate the expert predictions into a single prediction $\hat{Y}_t$ of $Y_t$. This is done using the well-studied Gibbs-aggregation approach with the soft*min* function (since *lower* loss implies *better* score here). Line 13 computes the aggregation weights $\bar{\pi}$; which are then packaged into a uniform row-stochastic matrix whose rows are given by $\bar{\pi}$. With these aggregation weights, our MoE-F algorithm generates a joint predictor in line 14. We associate $P$ to a Markov Q/intensity matrix $Q$. The matrix logarithm of $P$ is a viable candidate for $Q$, since $\exp(1 \cdot Q)$ would be a valid transition matrix; however, in general, the matrix logarithm of an arbitrary row-stochastic transition matrix is not a valid intensity matrix. Upon regularizing $P$ in line 18 to avoid pathologies, in line 19, we compute the best intensity matrix approximation of its matrix logarithm.

---

**Algorithm 1:** MoE-F Algorithm

1   Initialize $\pi$, $P$, and $Q$;
2   /* Step 1:   Optimal Parallel Filter   */
3   **For** $n = 1, \ldots, N$ **in parallel**
4      drift $\leftarrow Q_{t-1}^\top \pi_{t-1}^{(n)}$;
5      $\Delta L \leftarrow \ell(f^{(n)}(x_{[0,t)}), Y_t) - \ell(f^{(n)}(x_{[0,t-1)}), Y_{t-1})$;
6      $\Delta W \leftarrow (\Delta L - \bar{A}_{t-1}(\pi_{t-1}^{(n)}, Y_{[0,t-1]}))/B_{t-1}^{(n)}(Y_{t-1})$;
7      $\text{diff}_i \leftarrow (\pi^{(n:i)}(A_{t-1}(e_i, Y_{[0,t-1]}) - \bar{A}_{t-1}(\pi_{t-1}^{(n)}, Y_{[0,t-1]})))/B_{t-1}^{(n)}(Y_{t-1})$ ;
8      $\pi_t^{(n)} \leftarrow \pi_{t-1}^{(n)} + \text{drift} + \text{diff } \Delta W$;
9      $\hat{Y}_t^{(n)} \leftarrow (\pi_t^{(n)})^\top F(x_{[0,t)})$;
10      $s_n \leftarrow \ell(Y_t, \hat{Y}_t^{(n)})$ ;
11   **end**
12   /* Step 2:   Robust Aggregation     */
13   $\bar{\pi} \leftarrow \left(e^{-\lambda s_n}/(\sum_{i=1}^N e^{-\lambda s_i})\right)_{n=1}^N$;
14   $\hat{Y}_t \overset{\text{def.}}{=} \bar{\pi}^\top (\hat{Y}_t^{(n)})_{n=1}^N$;
15   **for** $n = 1, \ldots, N$ **do**
16      $P_{n,\cdot} \leftarrow \bar{\pi}$
17   **end for**
18   $P \leftarrow (1 - \alpha) P + \alpha I_N$;
19   $Q \leftarrow \text{ReLU}(\log(P)) - \text{diag}(\bar{1}_N^\top \text{ReLU}(\log(P)))$ ;
20   **return** *MoE-F Prediction* $\hat{Y}_t$;

---

## 4   THEORETICAL GUARANTEES

Our MoE-F algorithm revolves around the following set of *stochastic filtering equations*. Each filtering equation corresponds to the best estimate of a single expert on masking process $w.$ given only

the information available in their *running performance*. Interpreting "best estimate" in the $L^2$ sense, each seeks to predict the conditional distribution of $w.$ given the $\sigma$-algebra $\mathcal{F}_t^{(n)} \stackrel{\text{def.}}{=} \sigma\{\ell_s^{(n)}\}_{0 \le s \le t}$ generated by the running loss process $\ell_{\cdot}^{(n)} \stackrel{\text{def.}}{=} (\ell(Y_t, \hat{Y}_t^{(n)}))_{t \ge 0}$; i.e.

$$\pi_t^{(n)} \stackrel{\text{def.}}{=} \left(\mathbb{P}\big(w_t = e_i \mid \mathcal{F}_t^{(n)}\big)\right)_{i=1}^{N}.$$

An essential property of our filtering equations are that they provide a *closed-form* recursion for $\pi_{\cdot}^{(n)} \stackrel{\text{def.}}{=} (\pi_t^{(n)})_{t \ge 0}$ depending only on each of the expert models and on the accumulating observations from the target process $Y.$. This key property is rare in stochastic filtering paradigm, voiding the need for any Monte-Carlo simulation.

## 4.1 Guarantees for Step 1 - Online Parallel Filtering

Our main result comes in two variants, a binary classification and a regression variant. Both versions of our guarantees operate under the following regularity conditions of the path $x.$.

**Assumptions 4.1** (Regularity Conditions). *The path $x. : [0, \infty) \to \mathbb{R}^d$ is once continuously differentiable and there is a constant $C > 0$ such that, for each $t \ge 0$ and $i, j = 1, \ldots, d$, $(Q_t)_{i,j} \le C$. The loss function $\ell$ is either the binary cross entropy loss or the squared-norm loss.*

**Theorem 1** (Optimal Optimistic Prior for $n^{th}$ Expert). *Under Assumption 4.1, the best a posteriori estimate of the $n^{th}$ expert, $\pi_t^{(n)}$, satisfies the SDE*

$$\pi_t^{(n:i)} = \pi_0^i + \int_0^t (Q_s)_i^\top \pi_s^{(n)} \, ds + \int_0^t \frac{\pi_s^{(n:i)} \left(A_s(e_i, Y_{[0,s]}) - \bar{A}_s(\pi_s^{(n)}, Y_{[0,s]})\right)}{B_s(Y_{[0,s]})} d\overline{W}_u^{(n)}, \quad (5)$$

*where $(Q_t)_i$ denotes the $i^{th}$ row of the transitions matrix $Q_t$ at time $t \ge 0$, $\pi_0^i \stackrel{\text{def.}}{=} \mathbb{P}(w_0 = e_i)$. The "innovations process" $\overline{W}_{\cdot}^{(n)} \stackrel{\text{def.}}{=} (\overline{W}_t^{(n)})_{t \ge 0}$ is the following $(\mathbb{P}, \mathcal{F}_{\cdot}^n)$-Brownian motion*

$$\overline{W}_s^{(n)} \stackrel{\text{def.}}{=} \int_0^s \frac{dL_{[0,u]}^{(n)} - \bar{A}_u(Y_{[0,u]})}{B_u(Y_{[0,u]})} du, \tag{6}$$

*and each of the $n$ "running loss processes" $L_{\cdot}^{(n)}$ satisfy:*

  (i) **Classification:** *If $\ell$ is the binary cross entropy loss:*

$$dL_{[0,u]}^{(n)} = \frac{(Y_s - f^{(n)}(x_{[0,t)}))\Delta f^{(n)}(x_{[0,t)})}{(1 - f^{(n)}(x_{[0,t)}))f^{(n)}(x_{[0,t)})} + \log\left(\frac{f^{(n)}(x_{[0,t)})}{1 - f^{(n)}(x_{[0,t)})}\right)[w_t^\top F(x_{[0,t)})] \, dt + \log\left(\frac{f^{(n)}(x_{[0,t)})}{1 - f^{(n)}(x_{[0,t)})}\right) dW_t.$$

  (ii) **Regression:** *If $\ell$ is the squared-norm loss*

$$dL_{[0,u]}^{(n)} = 2(Y_t - f^{(n)}(x_{[0,t)}))\left([w_s^\top F(x_{[0,t)})] - \Delta f^{(n)}(x_{[0,t)}) + e^{t \ln(\delta^8)}\right) + 2(Y_t - f^{(n)}(x_{[0,t)}))e^{t \ln(\delta^4)} dW_t.$$

## 4.2 Guarantees for Step 2 - Robust Aggregation

The following bi-level optimality guarantee justifies the second step of our MoE-F algorithm.

**Theorem 2** (Bi-level Robust Updates to the $Q$-Matrix). *Let $\hat{Y}^{(1)}, \ldots, \hat{Y}^{(N)}, Y \in \mathbb{R}$ and $\ell : \mathbb{R}^2 \to \mathbb{R}$ be Borel measurable. Then $P \stackrel{\text{def.}}{=} [(\bar{\pi})_{n=1}^N]^\top$; where $\bar{\pi} \stackrel{\text{def.}}{=} \text{Softmin}\left(\lambda \left(\ell(Y^{(n)}, Y)\right)_{n=1}^N\right)$ minimizes*

$$\min_{P \in \mathcal{P}_N^U} \max_{m=1,\ldots,N} \underbrace{\sum_{n=1}^N P_{m,n} \, \ell(\hat{Y}_t^{(n)}, Y_t)}_{\text{Predictive Power of Ensemble}} + \frac{1}{\lambda} \underbrace{\sum_{n=1}^N P_{m,n} \, \log(P_{m,n}/N)}_{\text{Entropic Regularization}}. \tag{Inner}$$

*Let $\alpha \in (0, 1)$ then $P^\alpha \stackrel{\text{def.}}{=} (1 - \alpha)(\bar{\pi}, \ldots, \bar{\pi}) + \alpha I_N$ is a row stochastic matrix and, for $\alpha < 1$ large enough, $\log(P^\alpha)$ is well-defined. Furthermore, $Q \stackrel{\text{def.}}{=} \text{ReLU}\left(\log(P^\alpha)\right)$ is a minimizer of*

$$\min_{Q \in \mathcal{Q}} \|Q - \log(P_t^\alpha)\|_\infty. \tag{Outer}$$

We close our guarantees section by investigating the effect of regularizing the row stochastic matrix $P$ obtained in the inner-level optimization equation (Inner) to ensure its invertibility and the well-posedness of its matrix logarithm. We ask whether the probability distributions defined by the rows of $P_t^\alpha$ (whose state-space is $\{e_n\}_{n=1}^N$) depend continuously on $\alpha$. Indeed, the next result shows that the KL-divergence of the rows of $P_t^\alpha$ and those of $P_t$ (i.e. $\bar{\pi}$) are necessarily very close when $\alpha$ is small.

**Proposition 1** (Stability of Perturbations). *In the setting of Proposition 2, we have that*

$$\max_{i=1,\ldots,N} \mathrm{KL}(\bar{\pi}|(P_t^\alpha)_i) \le 2\alpha \left( -\frac{\log(\pi_{\min})}{1/\pi_{\min} - 1} - \frac{\log((1-\alpha)\,\pi_{\min})}{1/((1-\alpha)\,\pi_{\min}) - 1} \right)$$

*where $\pi_{\min} = \min_{i=1,\ldots,N} \bar{\pi}(> 0)$ and $(P_t^\alpha)_i$ denotes the $i^{th}$ row of $P_t^\alpha$.*

## 5 EXPERIMENTS

### 5.1 FINANCIAL MARKET MOVEMENT (FMM) TASK ON NIFTY USING MoE-F

We show the efficacy of our proposed MoE-F algorithm by using various SOTA class large language models on the *financial market movement* prediction task.

**Task.** The Financial Market Movement (FMM) prediction task for experts' evaluation can be defined as a ternary or binary market movement direction *classification task* among the labels' set $C = \{$*'Fall'*, *'Neutral'*, *'Rise'* $\}$ conditioned on history (or, expert memory) of window size $H$ (i.e., on the time window $[t - H + 1, t]$) – similar to the auto-regressive or causal generative language model (causal LM) training objective.

**Experts.** We consider a diverse list of SOTA general purpose instruction-tuned LLMs as experts for the experiments on our proposed MoE-F algorithm. For single LLM experts, we use Meta's open-weights models: Llama-2 (7B, 70B), Llama-3 (8B, 70B) [Touvron et al., 2023]. For mixture of experts (MoE) architecture models, we use SOTA open-weights models: Mixtral (8x7B) [Jiang et al., 2024] – which is a mixture of 8 Mistral (7B) [Jiang et al., 2023] models – and DBRX-Instruct [MosaicAI, 2024] with 132B total parameters and a mixture of 16 (fine-grained, smaller, 65x more combinations of) experts. For evaluation, we deployed these open-weights models as vLLM [Kwon et al., 2023] OpenAI compatible API endpoints and ran the dataset queries against them. We use API/model configurations like *guided-choice* and *max-tokens* to format class label converged expert responses alongside specific prompt instructions. We use the closed-source latest variants of the GPT-4 [OpenAI, 2023] class of models: GPT4o, using the OpenAI API. These experts are leading foundation models on current performance benchmarks on language understanding (MMLU [Hendrycks et al., 2021]), programming (HumanEval [Chen et al., 2021]), math (GSM8K [Cobbe et al., 2021]) tasks and other concurrent LLM benchmarks [Song et al., 2023, Liang et al., 2022].

**Datasets.** For real-world experiments on the defined FMM task, we use the US equities market movement (NYSE ticker: $SPY) dataset NIFTY ($\mathcal{D}_{LM}$) [Saqur et al., 2024]. The test split statistics which we used are recorded in Table 1.

Each sample of the $\mathcal{D}_{LM}$ contains high-quality, processed (one-turn) conversational queries for an expert instruction fine-tuned LLM, where a query, $x_q^t$, comprises of a prompt $x_p^t$ and a response $x_r^t$, i.e., $x_q^t = (x_p^t; x_r^t)$ corresponding to a day (or, time-step) $t$.

*Table 1:* Statistics of NIFTY test split

| Category | Statistics |
|---|---|
| Number of days ($T$) / increment ($\Delta t$) | **317** / 1 |
| Label support (Fall / Neutral / Rise) | 73 / 143 / 101 |
| Date range (*start* to *end*) | 2019-02-13 to 2020-09-21 |

For evaluation, at each time step $t$, an expert LLM is prompted ($x_p^t$) to predict the market movement the following day (i.e., $t + 1$), based on the market's current contextual information (relevant financial news headlines and the market's financial numerics – like the standard OHLCV and common technical indicators – from past few days capturing trends). Fig. 3 depicts a snapshot of an expert prompt $x_p^t$ for elucidation. Please see Fig. 6 in Appendix §C.1 for details.

For our ablative experiments, we utilise three additional datasets, namely StockNet aka. ACL18, BigData22 and CIKM datasets [Xu & Cohen, 2018, Soun et al., 2022, Wu et al., 2018]. These datasets have similar overarching (FMM) task design, but the prompted contextual information (e.g. social media opinions) and targeted asset classes (e.g. individual stock tickers, or price of gold) differ. We delegate the full details of these datasets in C.3 of Appendix §C Datasets.

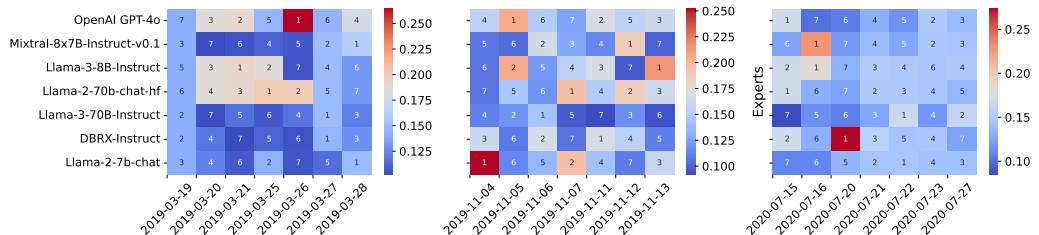

*Figure 3:* Example snapshot of the 'news' component on 2020-02-06, at the upstart of the global coronavirus epidemic (the text colors here convey negative and positive sentiments). An expert policy, $\pi_{LM}$'s prompt is composed of a task instruction as prefix, concatenated with the market context, and this news value concatenated: $s.t.\ x_p \leftarrow (x_{prefix}; x_{context}; x_{news})$.

*Figure 4:* Heatmap of expert weights and subsequent rankings for the sampled windows in Fig. 1.

*Table 2:* Results of applying **MoE-F** on seven SOTA experts on the NIFTY (*test split*). All values are mean of 3 (random) runs except GPT-4o. Best value in each metric row is in **bold**. Confusion matrices in appendix Fig. 7

| Metrics ↑ | LLM Experts | | | | | | | Experts Filter |
|---|---|---|---|---|---|---|---|---|
| | Llama-2 7b-chat | Llama-2 70b-chat | Llama-3 8B-Instruct | Llama-3 70B-Instruct | Mixtral-8x7B Instruct-v0.1 | DBRX Instruct | OpenAI GPT-4o | MoE-F (ours) |
| F1 | 0.22 | 0.33 | 0.35 | 0.20 | 0.34 | 0.34 | 0.34 | **0.52** |
| Acc | 0.27 | 0.37 | 0.39 | 0.30 | 0.33 | 0.34 | 0.37 | **0.57** |
| Precision | 0.35 | 0.33 | 0.31 | 0.32 | 0.36 | 0.36 | 0.33 | **0.61** |
| Recall | 0.27 | 0.37 | 0.39 | 0.30 | 0.33 | 0.34 | 0.37 | **0.57** |

### 5.1.1 RESULTS: MoE-F ON MIXTURE OF (MIXTURE OF) EXPERTS

Table 2 shows the results of running MoE-F on the collection of SOTA experts. While none of these remarkable experts out-performs others exceedingly or has an overall outstanding performance on the FMM task, filtering the expert decisions using MoE-F yields remarkable performance both in terms of overall task results and gains — with **17% real and 48.5% relative F1** measure **improvement** over the next best performing expert values ($\approx 35\%$). Fig. 1 depicts our MoE-F mechanism's decisions trajectory overlaid on the true market movement and expert decision trajectories.

**Ablations.** We use the Llama [Touvron et al., 2023] instruction-tuned, open-weights class of experts — specifically, Llama-2 7B, Llama-3 8B variants — to further analyse MoE-F at work. For ablation experiments, we create 4 additional expert variants (as standard LoRA [Hu et al., 2021] LLM adapters) by fine-tuning the base expert with the 4 stock movement datasets, namely: {`nifty`, `acl18`, `cikm18`, `bigdata22`}, and run them against the same task and evaluation test split from before. Table 3 shows the combined results of Llama 2 and 3 along with the 4 expert variants.

*Table 3:* Performance of Llama-2-7b-chat and Llama-3-8B-Instruct base models with (SFT LoRA adapter) variants on the NIFTY Stock Price Movement Prediction Task (*test* split).

| Metrics ↑ | Llama-2-7b-chat | | | | | | Llama-3-8B-Instruct | | | | | |
|---|---|---|---|---|---|---|---|---|---|---|---|---|
| | Base | +nifty | +acl18 | +bigdata22 | +cikm18 | MoE-F | Base | +nifty | +acl18 | +bigdata22 | +cikm18 | MoE-F |
| F1 Score | 0.22 | 0.28 | 0.20 | 0.29 | 0.27 | **0.43** | 0.34 | 0.36 | 0.19 | 0.23 | 0.24 | **0.43** |
| Accuracy | 0.27 | **0.45** | 0.25 | 0.29 | 0.27 | **0.45** | 0.39 | 0.41 | 0.26 | 0.26 | 0.28 | **0.47** |
| Precision | 0.35 | 0.20 | 0.36 | 0.32 | 0.31 | **0.44** | 0.31 | **0.56** | 0.44 | 0.28 | 0.31 | 0.48 |
| Recall | 0.27 | **0.45** | 0.25 | 0.29 | 0.27 | **0.45** | 0.39 | 0.41 | 0.26 | 0.26 | 0.28 | **0.47** |

*Table 4:* **Llama-2, 3** experts and MoE-F's performance decomposition by class labels on NIFTY test split.

| Expert | Fall (support: 73) | | | Neutral (support: 143) | | | Rise (support: 101) | | |
|---|---|---|---|---|---|---|---|---|---|
| | F1 | Precision | Recall | F1 | Precision | Recall | F1 | Precision | Recall |
| Llama-2-7b-chat | **0.35** | 0.23 | **0.75** | 0.19 | 0.40 | 0.13 | 0.17 | 0.34 | 0.11 |
| nifty | *0.00* | *0.00* | *0.00* | **0.62** | 0.45 | **1.00** | *0.00* | *0.00* | *0.00* |
| acl18 | 0.28 | 0.20 | 0.45 | 0.08 | 0.46 | 0.04 | **0.32** | 0.28 | **0.39** |
| cikm18 | 0.21 | 0.18 | 0.25 | 0.34 | 0.43 | 0.29 | 0.29 | 0.26 | 0.32 |
| bigdata22 | 0.21 | 0.19 | 0.23 | 0.31 | 0.42 | 0.24 | 0.31 | 0.26 | 0.38 |
| MoE-F | 0.30 | **0.27** | 0.33 | 0.60 | **0.54** | 0.66 | 0.30 | **0.43** | 0.23 |
| Llama-3-8B-Instruct | *0.00* | *0.00* | *0.00* | 0.49 | 0.46 | 0.51 | 0.38 | 0.31 | 0.49 |
| nifty | *0.00* | *0.00* | *0.00* | 0.48 | 0.46 | 0.51 | 0.42 | 0.35 | 0.54 |
| acl18 | **0.15** | 0.17 | **0.14** | 0.04 | 0.30 | 0.02 | 0.41 | 0.29 | **0.70** |
| cikm18 | 0.13 | 0.16 | 0.11 | 0.27 | 0.42 | 0.20 | 0.39 | 0.29 | 0.58 |
| bigdata22 | 0.14 | 0.17 | 0.12 | 0.20 | 0.38 | 0.13 | 0.40 | 0.29 | 0.62 |
| MoE-F | 0.12 | **0.45** | 0.07 | **0.56** | **0.56** | **0.57** | 0.48 | **0.39** | 0.62 |

**Discussions.** As before, using MoE-F delivers superior results for each of the two cases. However, we notice that the overall performance on the FMM task is lower than from earlier: 43% compared to 52% (Table 2). This aligns with our proposed theoretical performance guarantee relative to experts. Increasing the quality and quantity of experts with specialized capabilities improves MoE-F results. Intuitively, in presence of a vastly superior expert, MoE-F will levy higher weight on it for its decisions. If any under-performing (specialized) expert does better over a certain time-window (say during bearish market regime), it percolates up the decision weighting map accordingly.

**Decomposing Expert Performance by Class Labels.** Table 4 tabulates experts' performance by each of the three movement labels detailing another insight and verification of MoE-F's mechanism at work. While some experts have overall equivalent performance (e.g. the F1 score of '+*nifty*' and '+*bigdata22*' adapters), examining their label-specific results show these experts performance emanates from different decisions. To elucidate, the Llama-2 base expert was far better in predicting market 'Fall' than Llama-3, however, the latter handily outperforms when predicting 'Rise'. Similarly, we also see some experts are degenerate for some label predictions, like the '+nifty' expert that only tends to predict 'Neutral' (the imbalanced class label with highest support).

## 5.2 TIME-SERIES FORECASTING (TSF) USING MoE-F

The mainstream practice of using the classic Mean Squared Error (MSE) in the TSF field makes problems in the domain a suitable test-bed for our regression-based theorem (Theorem 4).

**Task.** Letting $L$ represent the length of some historical observation window, and $H$ the prediction horizon, a generic TSF problem can be formalized as

$$\bar{x}_{[t+1:t+H]} = f(x_{[t-L+1:t]}),$$

where $x_{[t-L+1:t]} \in \mathbb{R}^{L \times C}$ and $\bar{x}_{[t+1:t+H]} \in \mathbb{R}^{H \times C}$. Here, $C$ is the number of distinct features or channels of an agent's observation; the (MSE) loss measures the discrepancy between the predicted values $\bar{x}_{[t+1:t+H]}^{(i)}$ and the ground truth $y_{[t+1:t+H]}^{(i)}$ as $\mathcal{L} = \frac{1}{C} \sum_{i=1}^{C} \left\| y_{[t+1:t+H]}^{(i)} - \bar{x}_{[t+1:t+H]}^{(i)} \right\|_2^2$.

### 5.2.1 EXPERIMENTAL SETUP

**Experts and Datasets.** Table 5 presents a few, among a plethora of concurrent models in the busy TSF research area that come in a variety of flavours, from specialized TSF tasks only models like N-HiTS, N-BEATS [Challu et al., 2023, Oreshkin et al., 2019] to customized or fine-tuned LLMs like LLMTime [Gruver et al., 2024]. Immediately recent works in both these categories – like the SAMFormer [Ilbert et al., 2024] and MoiRai [Woo et al., 2024] – pushes the envelope on ideas and performance further. The goal of the TSF experiments here is not to achieve superior performance, but to show the application of our online MoE-F harness over any $N$ arbitrary expert models and effectively combine their strengths in an inexpensive, heuristic manner. Thus, for applying our filter, we chose three recent, well-adopted models with standardized implementation: DLinear, PatchTST, SparseTSF [Zeng et al., 2023, Nie et al., 2022, Lin et al., 2024] – using the author provided code from the latter to replicate the models' predictions over varying horizon $H$.

Similarly, we picked the most common and widely adopted datasets among a slew of mainstream TSF datasets (e.g. Electricity [Dua & Graff, 2017], Traffic [California Department of Transportation, 2023]) from concurrent literature in the LTSF domain: ETTh1&ETTh2 [Zhou et al., 2021]. Fig. 5 presents the summary of these TSF datasets.

*Figure 5:* Electricity Transformer.

| Datasets | ETTh1 & ETTh2 |
|---|---|
| Channels | 7 |
| Frequency | hourly |
| Timesteps | 17,420 |

**Results.** Table 5 presents MoE-F applied to a set of concurrent TSF experts. While our replicated performances of the bottom three experts, used for filtering, are lower than published results, the filtered predictions using MoE-F gains strong performance improvement using only three experts as-is and without any hyperparameters tuning (like optimal *lambda, alpha* values).

*Table 5:* MSE results of multivariate *Long-term Time-series Forecasting* comparing contemporary SOTA expert models and applying our proposed filtering harness MoE-F. Best results are highlighted in **bold**. Second best results underlined. Replicated performance results are presented using *4 decimal places*. Other performances quoted from respective papers. Only the bottom three models were used to generate MoE-F results.

| Dataset | ETTh1 | | | | ETTh2 | | | |
|---|---|---|---|---|---|---|---|---|
| **Horizon** | **96** | **192** | **336** | **720** | **96** | **192** | **336** | **720** |
| Informer [Zhou et al., 2021] | 0.865 | 1.008 | 1.107 | 1.181 | 3.755 | 5.602 | 4.721 | 3.647 |
| Autoformer [Wu et al., 2021] | 0.449 | 0.500 | 0.521 | 0.514 | 0.645 | 0.788 | 0.957 | 0.792 |
| Pyraformer [Liu et al., 2021] | 0.664 | 0.790 | 0.891 | 0.963 | 0.645 | 0.788 | 0.907 | 0.963 |
| FITS (2024) [Xu et al., 2024] | 0.375 | 0.408 | 0.429 | 0.427 | **0.274** | 0.333 | **0.340** | **0.374** |
| PatchTST [Nie et al., 2022] | 0.6897 | 0.6676 | 0.5997 | 0.6873 | 0.4273 | 0.4964 | 2.6450 | 0.4324 |
| DLinear [Zeng et al., 2023] | 0.3788 | 0.4212 | 0.4520 | 0.5230 | 0.2908 | 0.4091 | 0.5320 | 0.7430 |
| SparseTSF [Lin et al., 2024] | 0.3631 | **0.4000** | 0.4346 | 0.4238 | 0.2945 | 0.3399 | 0.3595 | 0.3831 |
| MoE-F (ours) | **0.3630** | 0.4165 | **0.4178** | **0.4157** | 0.2911 | **0.3294** | 0.4840 | 0.3917 |

## 6 RELATED WORK

**Closed-Form Finite Dimensional Filters.** Our proposed MoE-F utilizes finite-dimensional closed-form filtering equations via the Wonham-Shiryaev filter Wonham [1964], Širjaev [1965]. Such closed-form filtering equations rarely exist as the general stochastic filtering theory is infinite-dimensional Stratonovich [1959; 1960], Zakai [1969] and thus computationally intractable in practice. Other than the Wonham-Shiryaev filter, there is only a handful of such finite-dimensional filters with closed-form recursions. E.g. the Kalman-Bucy filter Kalman [1960], Bucy & Joseph [2005] or the Beneš filter Beneš [1981], both of which require highly rigid assumptions. Otherwise, one does typically have access to interacting systems of particles, so-called particle filters Djuric et al. [2003], Del Moral [2013], which are computationally intractable in high dimensions.

**Bayesian Mixture of Experts.** In the context of mixtures of experts, or large foundation models, one typically relies on a gating mechanism, or a learned routing among expert models Jacobs et al. [1991], Jordan & Jacobs [1994]. These gating mechanisms are often using a *static* Bayesian optimization approach via Gibbs posterior mixtures; e.g. Alquier [2008], Alquier et al. [2016], Andrychowicz et al. [2016], Rothfuss et al. [2021; 2023]. What the gating mechanisms involved in these approaches have in common is that they all are *static* in the sense that they do not learn in an online manner from dynamically arriving inputs and feedback. In contrast to these, our MoE-F model is an online/dynamic Bayesian optimization algorithm which dynamically generates posteriors using a different ($L^2$) notion of optimality rather than the extremal version used to define (static) Gibbs posterior mixtures.

**Learned Routing vs. Ours.** Most concurrent MoE of LLMs build on the idea of trainable/learned experts mixing, and use a routing layer Zhou et al. [2022] as introduced by Jacobs et al. [1991], Jordan & Jacobs [1994] in the 90s. Their efficacy was later shown in Shazeer et al. [2017] and numerous models and variants of this core idea have been successfully showcased in the current LLM context Fedus et al. [2022], Jiang et al. [2024], Dai et al. [2024], Gale et al. [2023]. Our work is orthogonal to this approach since this framework has no dependency on learned routing. Our online heuristic approach alongside with formal optimality guarantee allows online adaptation like adding, removing or hot-swapping experts on the fly.

## 7  CONCLUSION

We introduced a filtering-based gating mechanism for dynamically adapting mixing procedures to incoming data, unlike static classical MoE methods. Our algorithm enjoys optimality (Theorem 1) and robustness guarantees (Theorem 2) and effectively performs TSF in two orthogonal settings.

**Limitations & Future Work.**  Our numerical experiments focused solely on LLM experts, excluding random structures like neural SDEs Kidger et al. [2021], potentially missing technical market movements and stochastic effects. Future work will integrate both LLM and SDE-based experts to capture news-driven and technical market dynamics.

### ACKNOWLEDGMENTS

Anastasis Kratsios acknowledges financial support from an NSERC Discovery Grant No. RGPIN-2023-04482 and their McMaster Startup Funds. Raeid Saqur is supported by Canada NSERC CGS-D Doctoral Grant. The authors acknowledge that resources used in preparing this research were provided, in part, by the Province of Ontario, the Government of Canada through CIFAR, and companies sponsoring the Vector Institute https://vectorinstitute.ai/partnerships/current-partners/. The authors would like to thank Marshall Wang for helping with reference code for computing DBRX experiments.

### ETHICS STATEMENT

This research on stochastic filtering-based gating mechanisms for online MoE foundation models presents significant advancements, with broad plausible societal implications. Specifically, this technology can enhance adaptability in scientific predictions using complex dynamic systems, which is important in various disciplines including (but not limited to) financial markets, healthcare, and climate science for example. The ability to integrate multiple expert models dynamically can lead to more informed decision-making that is less constrained than conventional methods. However, as this technology is further refined, it will be important to prevent misuse in situations where advantage may be unfairly balanced, as in high-frequency trading. This technology should be used in transparent, accountable, and regularly audited frameworks to ensure its responsible deployment.

We note that access to highly-capable pre-trained foundation models is publicly available through open-weight LLM repositories, such as HuggingFace. The MoE-F filtering harness can be utilized to enhance the performance of many downstream tasks using an ensemble of off-the-shelf expert models, without the need for costly fine-tuning. However, there is a growing concern about the lack of transparency concerning the training data used in these models. Consequently, any inherent biases in the experts could be exacerbated by the filtering process. For example, a biased mortgage risk evaluating model can persistently rank higher due to predicting a majority/minority label.

### REPRODUCIBILITY STATEMENT

In our commitment to advancing open science and fostering transparency, we strive to make all aspects of our research fully accessible to the broader scientific community. To ensure the reproducibility of our work, we have included relevant datasets, detailed results, and links to code repositories (or anonymized links, where appropriate) as supplementary materials. These resources have been provided to facilitate the replication and verification of our findings, in alignment with the principles of the double-blind review process. [1]

---

[1]Supplementary material, including test results and scripts to replicate the results presented here, is available at: https://github.com/raeidsaqur/moe-f.

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

# Appendices

**Appendix Contents**

# A  THE MOE-F ALGORITHM

This section records a detailed version of Algorithm 1, which can also be rolled-forward online as the target process $Y_\cdot$ is dynamically observed.

---

**Algorithm 2:** The MoE-F Algorithm

---

**Input:** A time-horizon $T \in \mathbb{N}_+$, $N$ (pre-trained) experts $f^{(1)}, \ldots, f^{(N)}$, hyperparameters $\lambda > 0$, $\alpha \in (0,1)$ and $k \in \mathbb{N}_+$, target $(Y_t)_{t=0}^{T-1}$, and input signal $x_{[0:T-1]}$.

**Output:** A posterior Mixture Weights $w_t$

1   /* Initialize                                                       */

2   Initialize $\pi \stackrel{\text{def.}}{=} (\pi^{(n:i)})_{n,i=1}^{N} \leftarrow (1/N)_{n,i=1}^{N}$

3   $Q \leftarrow (1/(N-1)I_{i \neq j} - 1\,I_{i=j})_{i,j=1}^{N}$

4   $(L_-^{(n)})_{n=1}^{N} \leftarrow 0$

5   **for** $t = 0, \ldots, T-1$ **do**

6      **For** $n = 1, \ldots, N$ **in parallel**

7          $\bar{A} \leftarrow \bar{A}_t^{(n)}(\pi^{(n)}, Y_t)$

8          $B \leftarrow B_t^{(n)}(Y_t)\ \tilde{\pi} \leftarrow \pi$

9          $L^{(n)} \leftarrow \ell\big(f^{(n)}(x_{[0:t]})\big)$

10         $\Delta L \leftarrow L^{(n)} - L_-^{(n)}$

11         $\Delta \overline{W} \leftarrow \frac{\Delta L - \bar{A}}{B}$

12         $L_-^{(n)} \leftarrow L^{(n)}$

13         /* Update components of $n^{th}$ expert's posterior $(\pi^{(n)})$           */

14         **for** $i = 1, \ldots, N$ **do**

15             $A \leftarrow A_t^{(n)}(e_i, Y_t)$

16             $\text{drift} \leftarrow Q_i^\top \tilde{\pi}^{(n)}$

17             $\text{diffusion} \leftarrow \tilde{\pi}^{(n:i)}(A - \bar{A})/B$

18             $\pi^{(n:i)} \leftarrow \tilde{\pi}^{(n:i)} + \text{drift} + \text{diffusion}\Delta\overline{W}$

19         **end for**

20         $\pi^{(n)} \leftarrow \pi^{(n)} / \sum_i \pi^{(n:i)}$

21         $s_n \leftarrow \ell\big(Y_t, (\pi^{(n)})^\top F(x_{[0:t]})\big)$

22         $\hat{Y}_t^{(n)} \leftarrow (\pi^{(n)})^\top F(x_{[0:t]})$                    // Calculate expert scores

23      **end**

24      $\bar{\pi} \leftarrow \big(e^{-\lambda\,s_n} / \big(\sum_{i=1}^{N} e^{-\lambda\,s_i}\big)\big)_{n=1}^{N}$               // Get Expert Scores

25      $\hat{Y}_t \stackrel{\text{def.}}{=} \bar{\pi}^\top \big(\hat{Y}_t^{(n)}\big)_{n=1}^{N}$                    // Get time $t$ prediction

26      /* Update $Q$                                                       */

27      $\tilde{Q} \leftarrow Q$

28      **for** $n = 1, \ldots, N$ **do**

29         $P^{(n)} \leftarrow \bar{\pi}$

30      **end for**

31      $P \leftarrow (1-\alpha)\,P + \alpha I_N$

32      $Q \leftarrow \text{ReLU}(\log(P)) - \text{diag}(\bar{1}_N^\top \text{ReLU}(\log(P)))$

33   **end for**

34   **return** *Sequence of Mixture Predictions* $(\hat{Y}_t)_{t=0}^{T-1}$

---

## B  PROOFS

This section contains the proofs of our main result, generalizations thereof, and variants which apply to the quadratic (squared) loss. In the latter case, the necessary modifications to the algorithm and the overall proof structure are relatively similar but with key technical differences.

**Mild Generalizations and Further Discussion.**  We will consider the slightly more general case where the target process $Y.$ follows the generalized dynamics

$$
Y_t = Y_0 + \underbrace{\int_0^t w_t^\top F(x_s)\, ds}_{\text{Best Expert Estimate}} + \underbrace{\int_0^t \sigma_s\, dW_s}_{\text{(Generalized) Idiosyncratic Residual}}, \tag{7}
$$

where, there are constants $\alpha, C \geq 0$ with $C \leq 1$ such that: for each $t \geq 0$ one has $\sigma_t = C\, e^{-\alpha t}$. By the Itô-isometry, see [Cohen & Elliott, 2015, Lemma 12.1.4], we have that the variance of $\int_0^t \sigma_s\, dW_s$ is given by

$$
\varsigma_t^2 \overset{\text{def.}}{=} \mathbb{E}\left[\left(\int_0^t \sigma_s\, dW_s\right)^2\right] = \begin{cases} C^2\,(1 - e^{-\alpha 2t})/(2\alpha) & \text{if } \alpha > 0 \\ t & \text{if } \alpha = 0 \end{cases} \tag{8}
$$

Observe that, if $\alpha > 0$ then the variance of $\int_0^t \sigma_s\, dW_s$ asymptotically stabilizes at 1, as $t$ becomes arbitrarily large. In contrast, the variance of $\int_0^t \sigma_s\, dW_s$ diverges in the case where $\alpha = 0$ (which is the case considered in the main body of our paper).

**Intuition behind the choice of assumed fluctuations/diffusion.**  The intuition behind this modelling choice for the diffusion coefficient $\sigma.$ is based on ideas behind concentration of measure. Consider the case where $\alpha > 0$ in equation (8). Since we will be considering classification applications, then we will not want the idiosyncratic residual $\int_0^t \sigma_s W_s$ to push fluctuate outside the unit interval $[0, 1]$, or rather the probability that any fluctuation of $\int_0^t \sigma_s W_s$ is "large" should be small. Since $\int_0^t \sigma_s W_s$ has a Gaussian distribution, then note, by standard Gaussian concentration inequalities, we have that

$$
\mathbb{P}\left(\left|\int_0^t \sigma_s W_s\right| \geq 1/2\right) \leq e^{(1/2)^2/(2\varsigma_t^2)} = e^{-\alpha/\left(4C^2(1-e^{-\alpha 2t})\right)} \leq e^{-\alpha/(C^2 4)} \leq e^{-\alpha/4}. \tag{9}
$$

We can control the probability that any fluctuation is "large", meaning larger than $1/2$, by setting the right-hand side of equation (9) to be a prespecified "small" value $\delta \in (0, 1]$ and solving for the required $\alpha > 0$ parameter in terms of $\delta$ yields the specification $\alpha = \ln(1/\delta^4)$. If $d \geq 2$, then we may set $C = \frac{\sqrt{2}}{d}$ purely for convenience in simplifying expressions below.

In this case, for any hyperparameter $0 < \delta \leq 1$, the quantities in Theorem 1 become

$$
\pi_t^{(n:i)} = w_0^i + \int_0^t (Q_s)_i^\top \pi_s^{(n)}\, ds + \int_0^t \frac{\pi_s^{(n:i)}\left(A_s(e_i, Y_{[0,s]}) - \bar{A}_s(Y_{[0,s]})^\top \pi_s^{(n)}\right)}{B_s(Y_{[0,s]})}\, d\overline{W}_u^{(n)}, \tag{10}
$$

where $(Q_t)_i$ denotes the $i^{th}$ row of the transitions matrix $Q_t$ at time $t \geq 0$, $w_0^i \overset{\text{def.}}{=} \mathbb{P}(w_0 = e_i)$ and where the "innovations process" $\overline{W}.^{(n)} \overset{\text{def.}}{=} (\overline{W}_t^{(n)})_{t \geq 0}$ is the following $(\mathbb{P}, \mathcal{F}_.^n)$-Brownian motion

$$
\overline{W}_s^{(n)} \overset{\text{def.}}{=} \int_0^s \frac{dL_{[0,u]}^{(n)} - \bar{A}_u(Y_{[0,u]})}{B_u(Y_{[0,u]})}\, du,
$$

where the "stochastic differential" $dL_{[0,u]}^{(n)}$ is given by

$$
dL_{[0,u]}^{(n)} = \left(2\big(Y_u - f^{(n)}(x_u)\big)[\nabla_u f^{(n)}(x_u) + w_u^\top F(x_u)] - e^{-2t\ln(1/\delta^4)}\right) du
$$

$$
+ \frac{2^{3/2}}{d}\,(Y_u - f^{(n)}(x_u))e^{-t\ln(1/\delta^4)}\, dW_u.
$$

## B.1 PROOF OF THEOREM 1

We are now ready to state and prove two versions, one of which generalizes, our first main result (Theorem 1). We consider two cases. We obtain our main result by customizing Theorem 1 to our classification problem, where $D = 1$ and the range of each expert is in $\{0, 1\} \subset \mathbb{R}$, and setting $\sigma$ to be a specific constant in $(0, \infty)$. For convenience, if we postulate that $\sigma_t = 1$; i.e. it is a constant function of the path $y_{[0,t]}$ and of time $t \geq 0$. Note that, by the Itô-isometry, see [Cohen & Elliott, 2015, Lemma 12.1.4] the *idiosyncratic residual* term $\int_0^t \sigma_s(Y_{[0,s]})\, dW_s$ in equation (1) has a centred normal random distribution with variance $\int_0^t \sigma_s^2\, ds = t$.

### B.1.1 CASE I: BINARY CROSS-ENTROPY CASE

We now state and prove a mild generalization of Theorem 1.

**Theorem 3** (Optimal Optimistic Prior for $n^{th}$ Expert - Squared Loss Case). *Consider the binary cross-entropy loss*

$$\ell(\hat{y}, y) \overset{\text{def.}}{=} y \log(\hat{y}) + (1 - y) \log(1 - \hat{y})$$

*and fix a continuously differentiable path $x_. \in C^1(\mathbb{R})$.*

*Under Assumptions 4.1, the best a posteriori estimate of the $n^{th}$ expert, $\pi_t^{(n)}$, satisfies the following stochastic differential equation*

$$\pi_t^{(n:i)} = \pi_0^i + \int_0^t (Q_t)_i^\top \pi_s^{(n)}\, ds + \int_0^t \frac{\pi_s^{(n:i)}\left(A_s(e_i, Y_{[0,s]}) - \bar{A}_s(\pi^{(n)}, Y_{[0,s]})\right)}{B_s(Y_{[0,s]})} d\overline{W}_u^{(n)}, \quad (11)$$

*where $(Q_t)_i$ denotes the $i^{th}$ row of the transitions matrix $Q_t$ at time $t \geq 0$, $\pi_0^i \overset{\text{def.}}{=} \mathbb{P}(w_0 = e_i)$,*

$$A_t^{(n)}(w, y_{[0,t]}) \overset{\text{def.}}{=} -\frac{\left(Y_s - f^{(n)}(x_{[0,t)})\right)\Delta f^{(n)}(x_{[0,t)})}{\left(1 - f^{(n)}(x_{[0,t)})\right) f^{(n)}(x_{[0,t)})} - \log\left(\frac{f^{(n)}(x_{[0,t)})}{1 - f^{(n)}(x_{[0,t)})}\right)\left[w_s^\top F(x_{[0,s)})\right]$$

$$\bar{A}_t^{(n)}(\pi^{(n)}, y_{[0,t]}) \overset{\text{def.}}{=} \sum_{i=1}^d A_t(e_i, Y_{[0,t]})\,\pi^{(n:i)}$$

$$B_t^{(n)}(y_{[0,t]}) \overset{\text{def.}}{=} -\log\left(\frac{f^{(n)}(x_{[0,t)})}{1 - f^{(n)}(x_{[0,t)})}\right) e^{s\ln(\delta^4)}$$

$$F(x_{[0,t)}) \overset{\text{def.}}{=} \left(f^{(1)}(x_{[0,t)}), \dots, f^{(N)}(x_{[0,t)})\right)$$

*and the "innovations process" $\overline{W}_.^{(n)} \overset{\text{def.}}{=} (\overline{W}_t^{(n)})_{t \geq 0}$ is the following $(\mathbb{P}, \mathcal{F}_.^n)$-Brownian motion*

$$\overline{W}_s^{(n)} \overset{\text{def.}}{=} \int_0^s \frac{dL_{[0,u]}^{(n)} - \bar{A}_u(Y_{[0,u]})}{B_u(Y_{[0,u]})} du,$$

*where*

$$dL_{[0,u]}^{(n)} = d\ell(Y_t, \hat{f}^{(n)}(x_{[0,t)})) = \frac{\left(Y_s - f^{(n)}(x_{[0,t)})\right)\Delta f^{(n)}(x_{[0,t)})}{\left(1 - f^{(n)}(x_{[0,t)})\right) f^{(n)}(x_{[0,t)})} + \log\left(\frac{f^{(n)}(x_{[0,t)})}{1 - f^{(n)}(x_{[0,t)})}\right)\left[w_s^\top F(x_{[0,s)})\right]$$

$$+ \log\left(\frac{f^{(n)}(x_{[0,t)})}{1 - f^{(n)}(x_{[0,t)})}\right) e^{s\ln(\delta^4)} dW_s.$$

*Proof.* Fix a continuously differentiable path $x_. \in C^1(\mathbb{R})$. Define $\Delta f(x_{[0,t)}) \overset{\text{def.}}{=} \partial_t f(x_{[0,t)})$ and set $\ell : \mathbb{R} \ni y \mapsto (y - f(x_{[0,t)}))^2$.

First, observe that: for all $t \geq 0$, all $x. \in C^1(\mathbb{R})$ and all $y \in \mathbb{R}$ one has

$$
\begin{aligned}
\ell_t(y) &= - \left( y \log(f^{(n)}(x_{[0,t)})) + (1-y) \log(1 - f^{(n)}(x_{[0,t)})) \right) \\
\frac{\partial \ell_t}{\partial t}(y) &= - \frac{\left(y - f^{(n)}(x_{[0,t)})\right) \Delta f^{(n)}(x_{[0,t)})}{\left(1 - f^{(n)}(x_{[0,t)})\right) f^{(n)}(x_{[0,t)})}, \\
\frac{\partial \ell_t}{\partial y}(y) &= - \log \left( \frac{f^{(n)}(x_{[0,t)})}{1 - f^{(n)}(x_{[0,t)})} \right), \\
\frac{\partial^2 \ell_t}{\partial y^2}(y) &= 0.
\end{aligned}
\tag{12}
$$

Since $\ell \in C^\infty(\mathbb{R}^d \times \mathbb{R}^d)$, we assumed that the path $x. \in C^1([0,\infty), \mathbb{R}^d)$ then this, together with the postulated dynamics on $Y.^{(n)}$ imply that Itô's Lemma/Formula, see [Cohen & Elliott, 2015, Theorem 14.2.4], used on the map $\ell_t^{(n)} : [0,\infty) \times \mathbb{R}^D \ni (t,y) \rightarrow (y - f^{(n)}(x_t))^2 \in \mathbb{R}$ pre-composed with $Y.$ applies. Whence, our and the assumed dynamics on $Y.$, postulated in equation (1), imply that the process $L_t^{(n)} \stackrel{\text{def.}}{=} \ell_t^{(n)}(Y_t)$ satisfies the following stochastic differential equation

$$
\begin{aligned}
L_t^{(n)} = L_0^{(n)} &+ \int_0^t \frac{\partial \ell_s^{(n)}}{\partial s}(Y_s) \\
&+ \frac{\partial \ell_s^{(n)}}{\partial y}(Y_s) \left[ w_s^\top F(x_{[0,s)}) \right] \\
&+ \frac{1}{2} \frac{\partial^2 \ell_t^{(n)}}{\partial y^2}(Y_s) \, 2 e^{s \ln(\delta^8)} ds \\
&+ \int_0^t \frac{\partial \ell_t^{(n)}}{\partial y}(Y_s) e^{s \ln(\delta^4)} \, dW_s \\
= L_0^{(n)} &+ \int_0^t \frac{(-1)\left(Y_s - f^{(n)}(x_{[0,t)})\right) \Delta f^{(n)}(x_{[0,t)})}{\left(1 - f^{(n)}(x_{[0,t)})\right) f^{(n)}(x_{[0,t)})} \\
&+ (-1) \log \left( \frac{f^{(n)}(x_{[0,t)})}{1 - f^{(n)}(x_{[0,t)})} \right) \left[ w_s^\top F(x_{[0,s)}) \right] \\
&+ \frac{1}{2} 0 \ ds \\
&+ \int_0^t (-1) \log \left( \frac{f^{(n)}(x_{[0,t)})}{1 - f^{(n)}(x_{[0,t)})} \right) e^{s \ln(\delta^4)} \, dW_s
\end{aligned}
\tag{13}
$$

$$
\begin{aligned}
= L_0^{(n)} &+ \int_0^t -\frac{\left(Y_s - f^{(n)}(x_{[0,t)})\right) \Delta f^{(n)}(x_{[0,t)})}{\left(1 - f^{(n)}(x_{[0,t)})\right) f^{(n)}(x_{[0,t)})} - \log \left( \frac{f^{(n)}(x_{[0,t)})}{1 - f^{(n)}(x_{[0,t)})} \right) \left[ w_s^\top F(x_{[0,s)}) \right] \ ds \\
&- \log \left( \frac{f^{(n)}(x_{[0,t)})}{1 - f^{(n)}(x_{[0,t)})} \right) \int_0^t e^{s \ln(\delta^4)} \, dW_s
\end{aligned}
\tag{14}
$$

Synchronizing our notation with [Liptser & Shiryaev, 2001a, Equation (9.1)], we write

$$
\begin{aligned}
A_t(w, y_{[0,t]}) &\stackrel{\text{def.}}{=} - \frac{\left(Y_s - f^{(n)}(x_{[0,t)})\right) \Delta f^{(n)}(x_{[0,t)})}{\left(1 - f^{(n)}(x_{[0,t)})\right) f^{(n)}(x_{[0,t)})} - \log \left( \frac{f^{(n)}(x_{[0,t)})}{1 - f^{(n)}(x_{[0,t)})} \right) \left[ w_s^\top F(x_{[0,s)}) \right] \\
B_t(y_{[0,t]}) &\stackrel{\text{def.}}{=} - \log \left( \frac{f^{(n)}(x_{[0,t)})}{1 - f^{(n)}(x_{[0,t)})} \right) e^{s \ln(\delta^4)}
\end{aligned}
\tag{15}
$$

Under Assumptions 4.1, we may apply [Liptser & Shiryaev, 2001a, Theorem 9.1] to deduce that Then the a posteriori probability $\pi_t^{(n:i)} \stackrel{\text{def.}}{=} (\pi_t)_i$, satisfies a system of equations

$$
\pi_t^{(n:i)} = \pi_0^i + \int_0^t (Q_t)_i^\top \pi_s^{(n)} \, ds + \int_0^t \frac{\pi_s^{(n:i)} \left( A_s(e_i, Y_{[0,s]}) - \bar{A}_s(\pi^{(n)}, Y_{[0,s]}) \right)}{B_s(Y_{[0,s]})} d\overline{W}_u,
\tag{16}
$$

where $(Q_t)_i$ denotes the $i^{th}$ row of the $Q_t$/transitions matrix $Q_t$ at time $t \geq 0$, $\pi_0^i \stackrel{\text{def.}}{=} \mathbb{P}(w_0 = e_i)$, and the "innovations process" is the $(\mathbb{P}, \mathcal{F}^n_\cdot)$-Brownian motion given by

$$\overline{W}_s^{(n)} \stackrel{\text{def.}}{=} \int_0^s \frac{dL^{(n)}_{[0,u]} - \bar{A}_u(\pi^{(n)}, Y_{[0,u]})}{B_u(Y_{[0,u]})} du \tag{17}$$

and where

$$\bar{A}_t(\pi^{(n)}, y_{[0,t]}) \stackrel{\text{def.}}{=} \sum_{i=1}^d A_s(e_i, Y_{[0,t]})\, \pi^{(n:i)}.$$

This completes the proof. □

*Remark* 1. Setting $\delta = 1$ in the previous derivation yields the formulation of Theorem 1 found in the main body of the paper.

### B.1.2   CASE II: SQUARED LOSS

**Theorem 4** (Optimal Optimistic Prior for $n^{th}$ Expert - Squared Loss Case). *Let* $\ell(\hat{y}, y) \stackrel{\text{def.}}{=} (y - \hat{y})^2$ *and fix a continuously differentiable path* $x_\cdot \in C^1(\mathbb{R})$.

*Under Assumptions 4.1, the best a posteriori estimate of the $n^{th}$ expert, $\pi_t^{(n)}$, satisfies the following stochastic differential equation*

$$\pi_t^{(n:i)} = w_0^i + \int_0^t (Q_t)_i^\top \pi_s^{(n)}\, ds + \int_0^t \frac{\pi_s^{(n:i)} \left(A_s(e_i, Y_{[0,s]}) - \bar{A}_s(\pi^{(n)}, Y_{[0,s]})\right)}{B_s(Y_{[0,s]})} d\overline{W}_u^{(n)}, \tag{18}$$

*where $(Q_t)_i$ denotes the $i^{th}$ row of the transitions matrix $Q_t$ at time $t \geq 0$, $w_0^i \stackrel{\text{def.}}{=} \mathbb{P}(w_0 = e_i)$,*

$$A_t^{(n)}(w, y_{[0,t]}) \stackrel{\text{def.}}{=} 2\big(y_t - f^{(n)}(x_{[0,t)})\big)\big(w_t^\top F(x_{[0,t)}) - \Delta f^{(n)}(x_{[0,t)}) + e^{s \ln(\delta^8)}\big)$$

$$\bar{A}_t^{(n)}(\pi^{(n)}, y_{[0,t]}) \stackrel{\text{def.}}{=} \sum_{i=1}^d A_t(e_i, Y_{[0,t]})\, \pi^{(n:i)}$$

$$B_t^{(n)}(y_{[0,t]}) \stackrel{\text{def.}}{=} 2^{3/2}(y_t - f^{(n)}(x_{[0,t)}))e^{t \ln(\delta^4)}$$

$$F(x_{[0,t)}) \stackrel{\text{def.}}{=} \big(f^{(1)}(x_{[0,t)}), \ldots, f^{(N)}(x_{[0,t)})\big)$$

*and the "innovations process" $\overline{W}_\cdot^{(n)} \stackrel{\text{def.}}{=} (\overline{W}_t^{(n)})_{t \geq 0}$ is the following $(\mathbb{P}, \mathcal{F}^n_\cdot)$-Brownian motion*

$$\overline{W}_s^{(n)} \stackrel{\text{def.}}{=} \int_0^s \frac{dL^{(n)}_{[0,u]} - \bar{A}_u(Y_{[0,u]})}{B_u(Y_{[0,u]})} du,$$

*where*

$$\begin{aligned}
dL^{(n)}_{[0,u]} = d\ell(Y_t, \hat{f}^{(n)}(x_{[0,t)})) =\, & 2\big(Y_t - f^{(n)}(x_{[0,t)})\big)\big([w_s^\top F(x_{[0,t)})] - \Delta f^{(n)}(x_{[0,t)}) + e^{t \ln(\delta^8)}\big) \\
& + 2\big(Y_t - f^{(n)}(x_{[0,t)})\big)e^{t \ln(\delta^4)}\, dW_t.
\end{aligned}$$

*Proof.* Fix a continuously differentiable path $x_\cdot \in C^1(\mathbb{R})$. Define $\Delta f(x_{[0,t)}) \stackrel{\text{def.}}{=} \partial_t f(x_{[0,t)})$ and set $\ell : \mathbb{R} \ni y \mapsto (y - f(x_{[0,t)}))^2$.

First, observe that: for all $t \geq 0$, all $x_\cdot \in C^1(\mathbb{R})$ and all $y \in \mathbb{R}$ one has

$$\begin{aligned}
\ell_t^{(n)}(y) &= \big(y - f^{(n)}(x_{[0,t)})\big)^2 \\
\frac{\partial \ell_t}{\partial t}(y) &= 2\big(y - f^{(n)}(x_{[0,t)})\big)\big(-\Delta f^{(n)}(x_{[0,t)})\big), \\
\frac{\partial \ell_t}{\partial y}(y) &= 2\big(y - f^{(n)}(x_{[0,t)})\big), \\
\frac{\partial^2 \ell_t}{\partial y^2}(y) &= 2.
\end{aligned} \tag{19}$$

Since $\ell \in C^\infty(\mathbb{R}^d \times \mathbb{R}^d)$, we assumed that the path $x. \in C^1([0, \infty), \mathbb{R}^d)$ then this, together with the postulated dynamics on $Y_\cdot^{(n)}$ imply that Itô's Lemma/Formula, see [Cohen & Elliott, 2015, Theorem 14.2.4], used on the map $\ell_t^{(n)} : [0, \infty) \times \mathbb{R}^D \ni (t, y) \to (y - f^{(n)}(x_t))^2 \in \mathbb{R}$ pre-composed with $Y_\cdot$ applies. Whence, the computations in equation (19) and the assumed dynamics on $Y_\cdot$, postulated in equation (1), imply that the process $L_t^{(n)} \overset{\text{def.}}{=} \ell_t^{(n)}(Y_t)$ satisfies the following stochastic differential equation

$$
\begin{aligned}
L_t^{(n)} = L_0^{(n)} &+ \int_0^t \frac{\partial \ell_s^{(n)}}{\partial s}(Y_s) + \frac{\partial \ell_s^{(n)}}{\partial y}(Y_s)\left[w_s^\top F(x_{[0,s)})\right] + \frac{1}{2}\frac{\partial^2 \ell_t^{(n)}}{\partial y^2}(Y_s)\, 2e^{s\ln(\delta^8)}ds \\
&+ \int_0^t \frac{\partial \ell_t^{(n)}}{\partial y}(Y_s)e^{s\ln(\delta^4)}\, dW_s \\
= L_0^{(n)} &+ \int_0^t 2\big(Y_s - f^{(n)}(x_{[0,s)})\big)\big(\left[w_s^\top F(x_{[0,s)})\right] - \Delta f^{(n)}(x_{[0,s)}) + e^{s\ln(\delta^8)}\big)ds \qquad (20) \\
&+ \int_0^t 2\big(Y_s - f^{(n)}(x_{[0,s)})\big)\sqrt{2}e^{s\ln(\delta^4)}\, dW_s
\end{aligned}
$$

Synchronizing our notation with [Liptser & Shiryaev, 2001a, Equation (9.1)], we write

$$
\begin{aligned}
A_t(w, y_{[0,t]}) &\overset{\text{def.}}{=} 2\big(y_t - f^{(n)}(x_{[0,t)})\big)\big(w_t^\top F(x_{[0,t)}) - \Delta f^{(n)}(x_{[0,t)}) + e^{s\ln(\delta^8)}\big) \\
B_t(y_{[0,t]}) &\overset{\text{def.}}{=} 2^{3/2}(y_t - f^{(n)}(x_{[0,t)}))e^{t\ln(\delta^4)}
\end{aligned}
\qquad (21)
$$

Under Assumptions 4.1, we may apply [Liptser & Shiryaev, 2001a, Theorem 9.1] to deduce that Then the a posteriori probability $\pi_t^{n:0} \overset{\text{def.}}{=} (\pi_t)_0$, satisfies a system of equations

$$
\pi_t^{(n:i)} = w_0^i + \int_0^t (Q_t)_i^\top \pi_s^{(n)}\, ds + \int_0^t \frac{\pi_s^{(n:i)}\big(A_s(e_i, Y_{[0,s]}) - \bar{A}_s(\pi^{(n)}, Y_{[0,s]})\big)}{B_s(Y_{[0,s]})}d\overline{W}_u, \qquad (22)
$$

where $(Q_t)_i$ denotes the $i^{th}$ row of the $Q_t$/transitions matrix $Q_t$ at time $t \geq 0$, $w_0^i \overset{\text{def.}}{=} \mathbb{P}(w_0 = e_i)$, and the "innovations process" is the $(\mathbb{P}, \mathcal{F}_\cdot^n)$-Brownian motion given by

$$
\overline{W}_s^{(n)} \overset{\text{def.}}{=} \int_0^s \frac{dL_{[0,u]}^{(n)} - \bar{A}_u(\pi^{(n)}, Y_{[0,u]})}{B_u(Y_{[0,u]})}du \qquad (23)
$$

and where

$$
\bar{A}_t(\pi^{(n)}, y_{[0,t]}) \overset{\text{def.}}{=} \sum_{i=1}^d A_s(e_i, Y_{[0,t]})\, \pi^{(n:i)}.
$$

This completes the proof. $\qquad\qquad\qquad\qquad\qquad\qquad\qquad\qquad\qquad\qquad\qquad\qquad\qquad\qquad\square$

*Remark* 2. Setting $\delta = 1$ in the previous derivation yields the formulation of Theorem 1 found in the main body of the paper.

### B.2  PROOF OF THEOREM 2

The proof of Theorem 2 relies on the following result. Briefly, this result guarantees for the validity of the perturbation to the transition probability defined by

$$
P_t^\alpha \overset{\text{def.}}{=} (1-\alpha)P_t + \alpha I_N \qquad (24)
$$

for arbitrary $N \in \mathbb{N}_+$, $P_t \in \mathcal{P}_N^U$, $\alpha \in (0, 1)$, and where $I_N$ is the $N \times N$ identity matrix.

In what follows, we will use $\Delta_N \overset{\text{def.}}{=} \{w \in [0, 1]^N : \sum_{n=1}^N w_n\}$ to denote the probability $N$-simplex; which corresponds to the probability (measures) distributions supported on $N$ points. Here, these $N$ points are the experts themselves, and the probability of selecting any expert is interpreted as the relative credibility we ascribe to its historical predictive power.

**Proposition 2** (Regularity of Perturbations). *Let $N \in \mathbb{N}_+$, $\lambda > 0$, $s_1, \ldots, s_N \in \mathbb{R}$, $\bar{\pi} \in \Delta_N$ be given by*

$$\bar{\pi} \overset{\text{def.}}{=} \text{Softmin}\left(\lambda \left(s_n\right)_{n=1}^N\right) \text{ and } P_t \overset{\text{def.}}{=} [(\bar{\pi})_{n=1}^N]^N.$$

*For every $\alpha \in (0,1)$, the matrix $P_t^\alpha$ in equation (24), is invertible and (row) stochastic. If, moreover, all its real eigenvalues are non-negative, then $\log(P_t^\alpha)$ is well-defined and its rows sum to 0. In particular, setting $\alpha \geq 1 - 1/N$ guarantees that $\log(P_t^\alpha)$ exists, if $P_t^\alpha$ has real eigenvalues.*

We will now show our second main result, and the intermediate lemmata leading up to it. The next lemma states that if a (row) stochastic matrix is constructed by filling each of its rows with an element of the probability simplex, then shining it by an arbitrarily small abound and growing its diagonal proportionally yields a (row) stochastic matrix, which is necessarily invertible.

**Lemma 1** (Invertible Perturbations). *Let $N \in \mathbb{N}_+$ let $\pi \in \Delta_N$. If $P$ is a (row) stochastic matrix, then, for any $\alpha \in (0,1)$, the matrix $(1-\alpha)P + \alpha I_N$ is an invertible (row) stochastic matrix.*

*Proof of Lemma 1.* Let $\mathbf{1}_N \in \mathbb{R}^N$ be such that: for each $i = 1, \ldots, N$ we have $(\mathbf{1}_N)_i = 1$ (*i.e. $\mathbf{1}_N$ is a matrix of ones*). By construction $P = (\pi, \ldots, \pi)^\top$. Therefore, $P$ can be written as an outer product via

$$P = \mathbf{1}_N \pi^\top \tag{25}$$

Therefore, for any $\alpha \in (0,1)$, the perturbed matrix $(1-\alpha)P + \alpha I_N$ can be expressed as

$$(1-\alpha) P = \left((1-\alpha) \cdot \mathbf{1}_N\right) \pi^\top, \tag{26}$$

i.e. $(1-\alpha) P$ can be expressed as an outer product of vectors in $\mathbb{R}^N$; namely of $\left((1-\alpha) \cdot \mathbf{1}_N\right)$ and $\pi^\top$. Consequentially, our matrix of interest can be written as

$$(1-\alpha) P + \alpha I_N = \left((1-\alpha) \cdot \mathbf{1}_N\right) \pi^\top + \alpha I_N. \tag{27}$$

Note that, $\alpha I_N$ is invertible since $\det(\alpha I_N) = \alpha^N > 0$. Thus, the main result of Bartlett [1951] can be applied, which yields the condition: if $\alpha I_N$ is invertible (which it is) and if

$$1 + \mathbf{1}_N^\top \left(\alpha I_N\right)^{-1} \pi \neq 0 \tag{28}$$

then $\alpha I_N + \left((1-\alpha) \cdot \mathbf{1}_N\right) \pi^\top$ is invertible. Thus, we only need to verify that the condition holds in our case. Simplifying equation (28) yields

$$-1 \neq \mathbf{1}_N^\top \left(\alpha I_N\right)^{-1} \pi \tag{29}$$

$$= \frac{1}{\alpha} \mathbf{1}_N^\top \pi \tag{30}$$

$$= \frac{1}{\alpha} \sum_{n=1}^N \pi_n \tag{31}$$

$$= \frac{1}{\alpha} 1 = \frac{1}{\alpha}, \tag{32}$$

where equation (32) held since $\pi \in \Delta_N$. Consequentially, the identity in equation (27) and the computation in equation (29)-equation (32) imply that $(1-\alpha) P + \alpha I_N$ is invertible if $\alpha \neq -1$.

Finally, since $P$ is (row) stochastic and so is $I_N$ then, for each $i = 1, \ldots, N$, we have that

$$\sum_{j=1}^N \left((1-\alpha) P_i + \alpha I_N\right)_j = (1-\alpha) \sum_{j=1}^N P_{i,j} + \alpha 1 = (1-\alpha) 1 + \alpha = 1.$$

Whence, $(1-\alpha)P + \alpha I_N$ is (row) stochastic also. $\square$

We now provide a set of *sufficient* condition on $\alpha$, guaranteeing that the principal logarithm of $P_t^\alpha$ is well-defined. Furthermore, this lemme also shows that for $\alpha \in (0,1)$ large enough, as a function of $N$, the matrix $\log(P_t^\alpha)$ is necessarily a valid candidate for a Markov transition matrix (i.e. each of its rows sum to 0).

**Lemma 2** (Sufficient Condition for Existence). *If $\alpha \geq 1 - \frac{1}{N^2}$ then if either of the following holds:*

(i) **Real Case:** $P_t^\alpha$ has no complex eigenvalues,

(ii) **Complex Case:** $P$ is doubly stochastic (i.e. row and column stochastic),

then $\log(P_t^\alpha)$ exists and its rows sum to $0$.

*Proof of Lemma 2.* Since $P_t^\alpha$ is an $N \times N$ real (thus complex) matrix with real eigenvalues, then define

$$m \stackrel{\text{def.}}{=} \operatorname{tr}(P_t^\alpha)/N \text{ and } s^2 \stackrel{\text{def.}}{=} \operatorname{tr}((P_t^\alpha)^2)/N - m^2. \tag{33}$$

First, observe that since the entries of $P_t$ (in particular its diagonal elements) are all positive then

$$m = \operatorname{tr}(P_t^\alpha)/N \tag{34}$$

$$= \frac{1}{N} \sum_{i=1}^{N} \left((1-\alpha)\,\pi_i + \alpha\right) \tag{35}$$

$$= \frac{1}{N} \left((1-\alpha)\sum_{i=1}^{N} \pi_i + \alpha\sum_{i=1}^{N} 1\right) \tag{36}$$

$$= \frac{1}{N} \left((1-\alpha) + \alpha N\right) \tag{37}$$

$$= \frac{1}{N} \left(1 + \alpha(N-1)\right). \tag{38}$$

Next, we compute $s^2$. By Lemma 1, we have that $P_t^\alpha$ is a stochastic matrix and, therefore, so is its square (as the product of stochastic matrices is stochastic). Note that

$$\operatorname{tr}((P_t^\alpha)^2) \leq \max_{S \in \operatorname{Stoch}(N)} \operatorname{tr}(S) = \operatorname{tr}(I_N) = N \tag{39}$$

where $\operatorname{Stoch}(N)$ is the set of $N \times N$ stochastic matrices. Therefore, we bound $s^2$, defined in equation (33) using the "extremal trace bound" in equation (39) via

$$s^2 = \operatorname{tr}(P_t^\alpha)/N - m^2 \tag{40}$$

$$\leq N/N - m^2 \tag{41}$$

$$= 1 - \frac{1}{N^2}\left(1 + \alpha(N-1)\right)^2. \tag{42}$$

That is

$$-s \geq -\left(1 - \frac{1}{N^2}\left(1 + \alpha(N-1)\right)^2\right)^{1/2}.$$

Now, using lower-bound on the minimal eigenvalue of a square complex matrix with real eigenvalues using $m$ and $s$ in [Wolkowicz & Styan, 1980, Theorem 2.1] we have that

$$\lambda_{\min}(P_t^\alpha) \geq m - s(N-1)^{1/2} \tag{43}$$

$$\geq \frac{1}{N}\left(1 + \alpha(N-1)\right) - \left(1 - \frac{1}{N^2}\left(1 + \alpha(N-1)\right)^2\right)^{1/2}(N-1)^{1/2} \tag{44}$$

$$= \frac{1}{N}\left(1 + \alpha(N-1)\right) - \left(N^2 - \left(1 + \alpha(N-1)\right)^2\right)^{1/2}\frac{(N-1)^{1/2}}{N} \tag{45}$$

$$= \frac{1}{N}\left(\left(1 + \alpha(N-1)\right)\right. \tag{46}$$

$$\left. - \left[(N-1)\left((1-\alpha^2)N^2 + 2\alpha(\alpha-1)N - (\alpha-1)^2\right)\right]^{1/2}\right). \tag{47}$$

If $\alpha \geq 1 - \frac{1}{N^2}$ and $N > 1$ (which is always the case) then

$$\left(\left(1 + \alpha(N-1)\right) - \left[(N-1)\left((1-\alpha^2)N^2 + 2\alpha(\alpha-1)N - (\alpha-1)^2\right)\right]^{1/2}\right) > 0. \tag{48}$$

Therefore, equation (48) together with equation (46) imply that

$$\lambda_{\min}(P_t^\alpha) > 0$$

whenever $\alpha \geq 1 - \frac{1}{N^2}$.

Therefore, [Dunford & Schwartz, 1958, Theorem VII.1.10] implies that $\log(P_t^\alpha)$ exists, since $P_t^\alpha$ is a matrix whose spectrum does not contain $(-\infty, 0]$. Moreover, [Davies, 2010, Lemma 1] guarantees that the rows of $\log(P_t^\alpha)$ sum to 0.

Finally, we note that since $I_N$ is doubly stochastic then so is $P_t^\alpha$ provided that $P$ is. Therefore, $\bar{P}_t^\alpha$ is also a stochastic matrix and so is the product $\bar{P}_t^\alpha P_t^\alpha$ (as the product of (row) stochastic matrices is again a (row) stochastic matrix). Whence

$$s_a^2 \overset{\text{def.}}{=} \operatorname{tr}(\overline{P_t^\alpha} P_t^\alpha)/N - m^2 \leq 1 - \frac{1}{N^2}\left(1 + \alpha(N-1)\right)^2 \tag{49}$$

and the same argument may be applied with $s_a^2$ in place of $s^2$ upon using [Wolkowicz & Styan, 1980, Theorem 3.1] in place of [Wolkowicz & Styan, 1980, Theorem 2.1]; however, in this case we do not need to assume that the eigenvalues of $P_t^\alpha$ are real. In either case, this concludes our proof. $\square$

### B.2.1 COMPLETION OF THE PROOF OF THEOREM 2

*Proof of Theorem 2.* **Step 1 - Minimizer of Inner Problem equation (Inner):**
Since the elements of the set of $N \times N$ uniform stochastic matrices $\mathcal{P}_N^U$ all have identical rows then, $P$ is an optimizer of equation (Inner) if and only if its first row is a minimizer of

$$\min_{P \in \mathcal{P}_N^U} \sum_{n=1}^N P_{1,n}\,\ell(Y_t^{(n)}, Y_t) + \frac{1}{\lambda} \sum_{n=1}^N P_{1,n}\,\log(w_n/N). \tag{50}$$

Since the matrices in $\mathcal{P}_N^U$ are row-stochastic, then all their rows belong to the $N$ simplex $\Delta_N$. Therefore, $P$ is a minimizer of equation (50) if and only if its first row, which we denote by $\pi \overset{\text{def.}}{=} (P_{1,1}, \ldots, P_{1,N}) \in \Delta_N$ is a minimizer of

$$\min_{\pi \in \Delta_N} \sum_{n=1}^N \pi_j\,\ell(Y_t^{(n)}, Y_t) + \frac{1}{\lambda} \sum_{n=1}^N \pi_n\,\log(w_n/N). \tag{51}$$

Since the elements of $\Delta_N$ are in bijection with the set of probability measures on the $N$-point set $\{1, \ldots, N\}$, $\lambda > 0$, and $\sum_{n=1}^N \pi_n \log(w_n/N)$ is the KL-divergence (relative entropy) between the probability measure $\sum_{n=1}^N \pi_n \delta_n$ and the uniform measure $\sum_{n=1}^N \frac{1}{N} \delta_N$ (both on $\{1, \ldots, N\}$) then [Wang et al., 2020, Proposition 1] the unique minimizer of equation (51) is given by

$$\bar{\pi} \overset{\text{def.}}{=} \operatorname{Softmin}\left(\lambda\left(\ell(Y^{(n)}, Y)\right)_{n=1}^N\right).$$

Consequentially, the matrix $P \in \mathcal{P}_N^U$ whose rows are $\pi$ is a minimizer of equation (Inner).

**Step 2 - Minimizer of Outer Problem equation (Outer):**
By Proposition 2, for every $\alpha(0,1)$ the matrix

$$P^\alpha \overset{\text{def.}}{=} (1-\alpha)P + \alpha I_N$$

is row-stochastic and for $\alpha$ "large enough"; meaning for $\alpha \in (1 - 1/N, 1)$, the matrix $P^\alpha$ has all its eigenvalues in $(0, \infty)$. Therefore, by [Davies, 2010, Theorem 12] there is a minimizer of equation (Outer) and it is given in closed-form by

$$Q \overset{\text{def.}}{=} \operatorname{ReLU}\left(\log(P_t^\alpha)\right).$$

This concludes our proof. $\square$

*Proof of Proposition 2.* By Lemma 1, the matrix $P_t^\alpha$ is (row) stochastic and invertible. Thus, it has no zero-eigenvalues. If, moreover, the assumption holds that $P_t^\alpha$ has no negative eigenvalues then [Dunford & Schwartz, 1958, Theorem VII.1.10] guarantees that the (principle branch) of the matrix logarithm of $P_t^\alpha$ exists. Consequentially, [Davies, 2010, Lemma 1] applies from which we deduce that the rows of $\log(P_t^\alpha)$ sum to 0. The last claim follows directly from Lemma 2 (i). $\square$

### B.3 PROOF OF THE STABILITY GUARANTEE IN PROPOSITION 1

The following generalizes, thus implies, Proposition 1.

**Lemma 3** (Maximal KL Divergence for Perturbation in Lemma 1). *Let $\pi \in \Delta_N$, $\alpha \in [0,1)$, $i = 1, \ldots, N$, and let for each $i = 1, \ldots, N$ let $\pi^{\alpha,i} = (1-\alpha)\pi + \alpha\, e_i$ where $\{e_i\}_{i=1}^N$ is the standard basis of $\mathbb{R}^N$. If $p_{\min} \stackrel{\text{def.}}{=} \min_{i=1,\ldots,N} p_i > 0$ then*

$$\max_{i=1,\ldots,N} \mathrm{KL}(\pi|\pi^{\alpha,i}) \leq 2\alpha \left( -\frac{\log(p_{\min})}{1/p_{\min}-1} - \frac{\log((1-\alpha)\,p_{\min})}{1/((1-\alpha)\,p_{\min})-1} \right).$$

*Proof of Lemma 3.* By the (sharp) reverse Pinsker inequality in [Binette, 2019, Theorem 1], as formulated in [Binette, 2019, Example A], yields the bound

$$\mathrm{KL}(\pi|\pi^{\alpha,i}) \leq \mathrm{TV}(\pi|\pi^{\alpha,i}) \left( \frac{\log(1/p_{\min})}{1/p_{\min}-1} + \frac{\log(1/p_{\min}^{\alpha,i})}{1/p_{\min}^{\alpha,i}-1} \right) \tag{52}$$

where TV is the total variation distance between $\pi$ and $\pi^{\alpha,i}$, and $p_{\min}^{\alpha,i} = \min_{i=1,\ldots,N} \pi^{\alpha,i}$. By construction

$$(1-\alpha)p_{\min} \leq p_{\min}^{\alpha,i} \leq (1-\alpha)p_{\min} + \alpha.$$

Whence,

$$\frac{1}{p_{\min}^{\alpha,i}} \leq \frac{1}{(1-\alpha)p_{\min}} \quad \text{and} \quad \frac{1}{1/p_{\min}^{\alpha,i}-1} \leq \frac{1}{1/((1-\alpha)p_{\min}+\alpha)-1}. \tag{53}$$

Incorporating equation (53) into equation (52) yields

$$\begin{aligned}
\mathrm{KL}(\pi|\pi^{\alpha,i}) &\leq \mathrm{TV}(\pi, \pi^{\alpha,i}) \left( -\frac{\log(p_{\min})}{1/p_{\min}-1} - \frac{\log((1-\alpha)\,p_{\min})}{1/((1-\alpha)\,p_{\min})-1} \right) \\
&= \sum_{j=1}^N \left| \pi_j - \pi_j^{\alpha,i} \right| \left( -\frac{\log(p_{\min})}{1/p_{\min}-1} - \frac{\log((1-\alpha)\,p_{\min})}{1/((1-\alpha)\,p_{\min})-1} \right) \\
&= \sum_{j=1}^N \left| \alpha\pi_j + \alpha I_{j=i} \right| \left( -\frac{\log(p_{\min})}{1/p_{\min}-1} - \frac{\log((1-\alpha)\,p_{\min})}{1/((1-\alpha)\,p_{\min})-1} \right) \\
&\leq \left( \sum_{j=1}^N \left| \alpha\pi_j \right| + \alpha \right) \left( -\frac{\log(p_{\min})}{1/p_{\min}-1} - \frac{\log((1-\alpha)\,p_{\min})}{1/((1-\alpha)\,p_{\min})-1} \right) \\
&\leq \left( \alpha\sum_{j=1}^N \left| \pi_j \right| + \alpha \right) \left( -\frac{\log(p_{\min})}{1/p_{\min}-1} - \frac{\log((1-\alpha)\,p_{\min})}{1/((1-\alpha)\,p_{\min})-1} \right) \\
&\leq \left( \alpha\sum_{j=1}^N \pi_j + \alpha \right) \left( -\frac{\log(p_{\min})}{1/p_{\min}-1} - \frac{\log((1-\alpha)\,p_{\min})}{1/((1-\alpha)\,p_{\min})-1} \right) \tag{54} \\
&\leq \left( \alpha + \alpha \right) \left( -\frac{\log(p_{\min})}{1/p_{\min}-1} - \frac{\log((1-\alpha)\,p_{\min})}{1/((1-\alpha)\,p_{\min})-1} \right) \tag{55} \\
&= 2\alpha \left( -\frac{\log(p_{\min})}{1/p_{\min}-1} - \frac{\log((1-\alpha)\,p_{\min})}{1/((1-\alpha)\,p_{\min})-1} \right).
\end{aligned}$$

where equation (54) held since $\pi \in \Delta_N$ and therefore, $\pi_i \geq 0$ for each $i = 1, \ldots, N$, and equation (55) held since $\sum_{i=1}^N \pi_i = 1$ again due to the fact that $\pi \in \Delta_N$. $\square$

## C    DATASETS AND BENCHMARKS

### C.1    NIFTY DATASET

The **N**ews-**I**nformed **F**inancial **T**rend **Y**ield (NIFTY) dataset Saqur et al. [2024] is a processed and curated daily news headlines dataset for the stock (US Equities) market price movement prediction task. NIFTY is comprised of two related datasets, NIFTY-LM and NIFTY-RL. In this section we outline the composition of the two datasets, and comment on additional details.

**Dataset statistics .**    Table 6 and Table 7 present pertinent statistics related to the dataset.

<table>
<tr><td colspan="2">Table 6: Statistics and breakdown of splits sizes</td></tr>
<tr><th>Category</th><th>Statistics</th></tr>
<tr><td>Number of data points</td><td>2111</td></tr>
<tr><td>Number of Rise/Fall/Neutral label</td><td>558 / 433 / 1122</td></tr>
<tr><td>Train/Test/Evaluation split</td><td>1477 / 317 / 317</td></tr>
</table>

<table>
<tr><td colspan="3">Table 7: Date Ranges of news headlines in splits</td></tr>
<tr><th>Split</th><th>Num. Samples</th><th>Date range</th></tr>
<tr><td>Train</td><td>1477</td><td>2010-01-06 to 2017-06-27</td></tr>
<tr><td>Valid</td><td>317</td><td>2017-06-28 to 2019-02-12</td></tr>
<tr><td>Test</td><td>317</td><td>2019-02-13 to 2020-09-21</td></tr>
</table>

*(a)* Instruction component of a $\pi_{LM}$ policy query $x_q$.

*(b)* The market's **history** is provided as the past $t$ days of numerical statistics like the (OHLCV) price (in blue) and common technical indicators (in orange) (e.g. moving averages) data.

*Figure 6:* Breaking down the instruction or prompt prefix, and market context components of a prompt, $x_p$.

### C.1.1    NIFTY-LM: SFT FINE-TUNING DATASET

The NIFTY-LM prompt dataset was created to finetune and evaluate LLMs on predicting future stock movement given previous market data and news headlines. The dataset was assembled by aggregating information from three distinct sources from January 6, 2010, to September 21, 2020. The compilation includes headlines from The **Wall Street Journal** and **Reuters News**, as well as market data of the $SPY index from **Yahoo Finance**. The NIFTY-LM dataset consists of:

- **Meta data**: Dates and data ID.
- **Prompt** ($x_p$): LLM question ($x_{question}$), market data from previous days ($x_{context}$), and news headlines ($x_{news}$).
- **Response**: Qualitative movement label ($x_r$) $\in \{Rise, Fall, Neutral\}$, and percentage change of the closing price of the $SPY index.

To generate LLM questions, ($\boldsymbol{x_{question}}$), the authors used the self-instruct Wang et al. [2023] framework and OpenAI GPT4 to create 20 synthetic variations of the instruction below:

> Create 20 variations of the instruction below.
> Examine the given market information and news headlines data on DATE to forecast whether the $SPY index will rise, fall, or remain unchanged. If you think the movement will be less than 0.5%, then return 'Neutral'. Respond with Rise, Fall, or Neutral and your reasoning in a new paragraph.

Where `DATE` would be substituted later, during the training phase with a corresponding date.

**Context.** The key 'context' ($x_{context}$) was constructed to have newline delimited market metrics over the past T ($\approx 10$) days (N.B. Not all market data for the past days for were available and therefore prompts might have less than 10 days of market metrics.).

Table 8 show the details of financial context provided in each day's sample.

*Table 8:* Summary of the dataset columns with their respective descriptions.

| Column Name | Description |
|---|---|
| Date | Date of the trading session |
| Opening Price | Stock's opening market price |
| Daily High | Highest trading price of the day |
| Daily Low | Lowest trading price of the day |
| Closing Price | Stock's closing market price |
| Adjusted Closing Price | Closing price adjusted for splits and dividends |
| Volume | Total shares traded during the day |
| Percentage Change | Day-over-day percentage change in closing price |
| MACD | Momentum indicator showing the relationship between two moving averages |
| Bollinger Upper Band | Upper boundary of the Bollinger Bands, set at two standard deviations above the average |
| Bollinger Lower Band | Lower boundary, set at two standard deviations below the average |
| 30-Day RSI | Momentum oscillator measuring speed and change of price movements |
| 30-Day CCI | Indicator identifying cyclical trends over 30 days |
| 30-Day DX | Indicates the strength of price trends over 30 days |
| 30-Day SMA | Average closing price over the past 30 days |
| 60-Day SMA | Average closing price over the past 60 days |

**News Headlines.** ($x_{news}$): Final list of filtered headlines from the aggregation pipeline. The non-finance related headlines were filtered out by performing a similarity search with SBERT model, "all-MiniLM-L6-v2" Reimers & Gurevych [2019]. Each headline was compared to a set of artificially generated financial headlines generated by GPT-4, with the prompt *"Generate 20 financial news headlines"*. Headlines with a similarity score below 0.2, were excluded from the dataset. To respect the prompting 'context length' of LLMs, in instances where the prompt exceeded a length of 3000 words, a further refinement process was employed. This process involved the elimination of words with a tf-idf Sammut & Webb [2010] score below 0.2 and truncating the prompt to a maximum of 3000 words.

It is also important to note that the dataset does not encompass all calendar dates within the specified time range. This limitation emanates from the trading calendar days, and absence of relevant financial news headlines for certain dates.

**Label.** ($x_r$): The label is determined by the percentage change in closing prices from one day to the next, as defined in equation 56. This percentage change is categorized into three labels: {Rise, Fall, Neutral}, based on the thresholds specified in equation 57.

$$PCT_{\text{change}} = \left( \frac{\text{Closing Price}_t - \text{Closing Price}_{t-1}}{\text{Closing Price}_{t-1}} \right) \times 100\% \tag{56}$$

$$x_r = \begin{cases} \text{Fall} & \text{if } PCT_{\text{change}} < -0.5\% \\ \text{Neutral} & \text{if } -0.5\% \leq PCT_{\text{change}} \leq 0.5\% \\ \text{Rise} & \text{if } PCT_{\text{change}} > 0.5\% \end{cases} \tag{57}$$

## C.2 NIFTY-RL: PREFERENCES DATASET

The preference dataset is a variation of the fine-tuning dataset and it is designed for alignment training of LLMs using reward model. In NIFTY-RL, labels are omitted and replaced with chosen and rejected

results. The chosen result is a label corresponding to a rise, a fall or neutral movement in the stock market and is equivalent to the response in NIFTY-LM. The rejected result is a random label not equal to the chosen label.

- **Metadata**: Includes dates and data identifiers.
- **Prompt** ($x_p$): Includes an LLM instruction ($x_{question}$), preceding market data ($x_{context}$), and relevant news headlines ($x_{news}$).
- **Chosen Result**: A qualitative movement label ($x_r$) from $\{Rise, Fall, Neutral\}$ indicating the predicted market trend.
- **Rejected Result**: A label ($\overline{x}_r$) randomly selected from $\{Rise, Fall, Neutral, Surrender\}\setminus\{x_r\}$, representing an incorrect market prediction.

### C.3 FLARE BENCHMARK DATASETS

**Stock Movement Prediction Datasets and Tasks: Flare-SM tasks.** **FLARE** proposed by Xie et al. [2023], extends to include one financial prediction task – the **CIKM** dataset Wu et al. [2018] as an evaluation task among (four) other general financial NLP tasks. Under the hood, this benchmark is a fork of the '*lm-eval*' harness Gao et al. [2021] with addendums. Other stock price movement prediction from social dataset include what is referred to as *ACL18* (or, 'acl18') in this paper is essentially the **StockNet** Xu & Cohen [2018] dataset which comprises of stock tweets of 88 stock tickers from 9 financial market industries from Twitter over two years (from 2014-2015) aligned with their corresponding historical price data. **BigData22** [Soun et al., 2022] is another more recent tweets dataset comprising of tweets about 50 stock tickers during the period 2019-07-05 to 2020-06-30.

*Table 9:* Summary of Flare stock price movement datasets. The 'Stocks' column indicates the total number of different stock tickers referenced. The 'Tweets' and 'Days' columns represent the number of tweets and days respectively in each dataset.

| Data | Stocks | Tweets | Days | Start Date | End Date |
|------|--------|--------|------|------------|----------|
| ACL18 | 87 | 106,271 | 696 | 2014-01-02 | 2015-12-30 |
| BigData22 | 50 | 272,762 | 362 | 2019-07-05 | 2020-06-30 |
| CIKM18 | 38 | 955,788 | 352 | 2017-01-03 | 2017-12-28 |

## D ADDITIONAL BACKGROUND MATERIAL

In an effort to keep our paper as self-contained as possible, this section contains additional background material used in our derivations and in the formulations of our technical results.

### D.1 MATRIX LOGARITHMS

The (principal) logarithm of an $N \times N$ matrix $A$ whose spectrum does not contain $(-\infty, 0]$ in $\mathbb{C}$ is defined by

$$\log(A) \stackrel{\text{def.}}{=} \frac{1}{2\pi i} \int_\gamma \log(z) \, (zI_N - A)^{-1} \, dz$$

where $\log(z)$ is the principal logarithm of $z$ (in the complex plane) and $\gamma$ is a closed curve in $\mathbb{C} \setminus (-\infty, 0]$ containing the eigenspectrum of $A$.

### D.2 THE SHIRAYEV-WONHAM FILTER

To keep our paper as self-contained as possible, we included some brief background on *stochastic filtering*. Namely, this appendix contains background material on the Shirayev-Wonham (stochastic) filter, studied in [Liptser & Shiryaev, 2001b, Chapter 9].

Consider a complete probability space $(\Omega, \mathcal{F}, P)$ equipped with a non-decreasing sequence of right-continuous sub-$\sigma$-algebras $\mathcal{F}_t, 0 \le t \le T$. Let $\theta = (\theta_t, \mathcal{F}_t), 0 \le t \le T$, denote a real right-continuous Markov process taking values in the countable set $E = \{\alpha, \beta, \gamma, \ldots\}$. Additionally,

let $W = (W_t, \mathcal{F}_t)$, $0 \le t \le T$, be a standard Wiener process independent of $\theta$, and let $\xi_0$ be a $\mathcal{F}_0$-measurable random variable independent of $\theta$. We assume the existence of nonanticipative functionals $A_t(\epsilon, x)$ and $B_t(x)$ that define

$$d\xi_t = A_t(\theta_t, \xi)dt + B_t(\xi)dW_t \tag{58}$$

and satisfy the following conditions.

$$A_t^2(\epsilon_t, x) \le L_1 \int_0^t (1 + x_s^2)dK(s) + L_2(1 + \epsilon_t^2 + x_t^2), \tag{59}$$

$$0 < C \le B_t^2(x) \le L_1 \int_0^t (1 + x_s^2)dK(s) + L_2(1 + x_t^2), \tag{60}$$

$$|A_t(\epsilon_t, x) - A_t(\epsilon_t, y)|^2 + |B_t(x) - B_t(y)|^2 \le L_1 \int_0^t (x_s - y_s)^2 dK(s) + L_2(x_t - y_t)^2, \tag{61}$$

where $C, L_1, L_2$ are certain constants, $K(s)$ is a non-decreasing right continuous function, $0 \le K(s) \le 1, x \in C_T, y \in C_T, \epsilon_t \in E, 0 \le t \le T$.

Along with Equations (59) to (61) it will also be assumed that

$$M\xi_0^2 < \infty, \tag{62}$$

and

$$M\int_0^T \theta_t^2 dt < \infty. \tag{63}$$

Define

$$p_\beta(t) \stackrel{\text{def.}}{=} P(\theta_t = \beta),$$

$$p_{\beta\alpha}(t, s) \stackrel{\text{def.}}{=} P(\theta_t = \beta | \theta_s = \alpha), \quad 0 \le s < t \le T, \quad \beta, \alpha \in E,$$

and assume there exist a function $\lambda_{\alpha\beta}(t), 0 \le t \le T, \alpha, \beta \in E$, that is

$$\text{continuous over } t, \text{ (uniformly over } \alpha, \beta) \tag{64}$$

$$|\lambda_{\alpha\beta}(t)| \le K \tag{65}$$

$$|p_{\beta\alpha}(t + \Delta, t) - \delta(\beta, \alpha) - \lambda_{\alpha\beta}(t) \cdot \Delta| \le o(\Delta), \tag{66}$$

where $\delta(\beta, \alpha)$ is a Kronecker's symbol and the value $o(\Delta)/\Delta \to 0$ as $\Delta \to 0$ (uniformly over $\alpha, \beta$).

Let the Equations (59) to (66) be fulfilled. Then the a posteriori probability $\pi_\beta(t), \beta \in \mathcal{E}$, satisfies a system of equations

$$\pi_\beta(t) = p_\beta(0) + \int_0^t \mathcal{L}^* \pi_\beta(u)du + \int_0^t \pi_\beta(u)\frac{A_u(\beta, \xi) - \bar{A}_u(\xi)}{B_u(\xi)}d\overline{W}_u,$$

where

$$\mathcal{L}^* \pi_\beta(u) = \sum_{\gamma \in \mathcal{E}} \lambda_{\gamma\beta}(u)\pi_\gamma(u) \quad \text{and} \quad \bar{A}_u(\xi) = \sum_{\gamma \in \mathcal{E}} A_u(\gamma, \xi)\pi_\gamma(u),$$

and $\overline{W} = (\overline{W}_t, \mathcal{F}_t)$ is a Wiener process with

$$\overline{W}_t = \int_0^t \frac{d\xi_u - \bar{A}_u(\xi)}{B_u(\xi)}du.$$

# E    ADDITIONAL DISCUSSIONS

We have updated the anonymous git repo with all the experiment results and added script to easily replicate the main results of the paper in Table 2.

## E.1    F1-MEASURE IN FMM TASK

For evaluation of model performances on the financial market movement (FMM) task using ternary class labels, we use the scikit-learn library methods. For multi-class, $n$, labels (with $n > 2$), the choice of *averaging* is important. The tabulated results were evaluated with default averaging set to "*weighted*" – where metrics for each label were calculated, then average weighted by their corresponding support (the number of true instances for each label). This alters the global averaging 'macro' (equal weight) for label imbalance. Imbalanced label support can result in an F-score that is lower or not in between precision and recall.

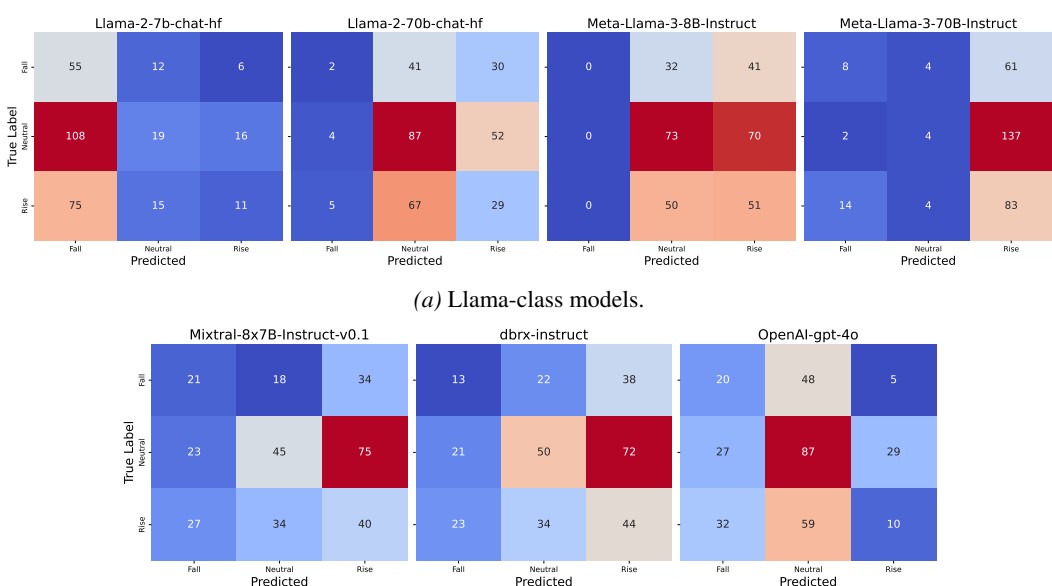

*(a)* Llama-class models.

*(b)* Mixture-of-experts class models and the state-of-the-art GPT-4 model.

*Figure 7:* Confusion matrices for Table 2. The first row highlights the Llama-class models, and the second row focuses on mixture-of-experts and GPT-4 models.

## E.2    TIME-SERIES FORECASTING EXPERIMENTS: ADDITIONAL DETAILS

This section provides additional discussion in support of the long-horizon time-series forecasting (LTSF) experiment covered in §5.2.

### E.2.1    FORECASTING GRANULARITY IN TIME-SERIES FORCASTING: IMS VS. DMS

In LTSF works, the decoding granularity is dichotomized in the following two categories:

**I) Iterated/Incremental Multi-step (IMS).**  : This is auto-regressive language-modeling or generative style prediction decoding where each step in prediction horizon $H$ is iteratively predicted: $\hat{x}_{t+1}$, and is used for the prediction for the subsequent time-step. The common limitations of such forecasting granularity is 'error accumulation' over time as the decoder builds on the error from previous steps while iteratively making subsequent predictions. Additionally, the run-time complexity is to the order of the length of the horizon $H$.

**II) Direct Multi-step (DMS).**  : As the name implies, in this decoding or forecasting approach, the entire forecasting horizon $H$ is predicted at one go: $\hat{x}_{t+1:t+H}$. While computationally more attractive

(much faster than IMS), this approach may not incorporate seasonality/periodicity in the time-series. Modern TSF specialist models, especially the transformer-based TSF architectures, tend to follow this scheme to avoid the quadratic cost (to the length of input) associated with attention.

### E.2.2    CHANNEL INDEPENDENT STRATEGY

Recent advancements in Long-Term Series Forecasting (LTSF) have increasingly embraced a **Channel Independent (CI) approach** for handling multivariate time series data Han et al. [2024]. The CI strategy simplifies forecasting by isolating each (channel or feature as) univariate time series within the dataset, allowing the model to focus on predicting individual channels independently. Unlike traditional methods that leverage the entire multivariate historical data to make forecasts, the CI approach seeks a shared function $f : x^{(i)}_{t-L+1:t} \in \mathbb{R}^L \rightarrow \hat{x}^{(i)}_{t+1:t+H} \in \mathbb{R}^H$ for each univariate series, providing a streamlined model for each channel and reducing the need to account for inter-channel dependencies.

### E.2.3    OUR SETUP

For our experiments, the historical observation window (*aka. look-back window* or *lag period*), $L$, is kept constant at 720 time-steps to be consistent and comparable with the literature. We follow the channel-independent strategy similar to the three expert models used for filtering - giving us $C$ number of distinct features or channels of an agent's observation. The (MSE) loss is then measured as the discrepancy between the predicted values $\bar{x}^{(i)}_{t+1:t+H}$ and the ground truth $y^{(i)}_{t+1:t+H}$ as

$$\mathcal{L} = \frac{1}{C} \sum_{i=1}^{C} \left\| y^{(i)}_{t+1:t+H} - \bar{x}^{(i)}_{t+1:t+H} \right\|_2^2. \tag{67}$$

