# OpenReview forum: "Filtered not Mixed: Filtering-Based Online Gating for Mixture of Large Language Models"
_ICLR.cc/2025/Conference — ICLR 2025 Poster_

### Official Review · Reviewer_bkmt · 2024-10-31

**Soundness:** 3
**Presentation:** 2
**Contribution:** 3
**Rating:** 6
**Confidence:** 3

**Summary:**

This paper introduces MoE-F, a novel mechanism for adaptively combining multiple pre-trained Large Language Models (LLMs) for online time-series prediction tasks. The key innovation is framing expert selection as a finite state-space continuous-time Hidden Markov model and leveraging stochastic filtering techniques to dynamically weight expert predictions.

**Strengths:**

The paper demonstrates significant originality in its approach to combining LLMs through the lens of stochastic filtering theory. This represents a creative fusion of classical control theory with modern machine learning. The theoretical foundations are rigorous, with formal optimality guarantees derived for both the parallel filtering and robust aggregation phases.

**Weaknesses:**

While the theoretical analysis is strong, there are some limitations in the experimental validation. The financial market experiments focus primarily on a single dataset (NIFTY), and broader validation across different types of time series problems would strengthen the claims of generality. The ablation studies, while informative, could be expanded to provide deeper insights into the relative importance of different components of the MoE-F architecture.

**Questions:**

How does the performance of MoE-F degrade with increasing number of experts, and what are the computational scaling implications?

What is the impact of different market regimes (high volatility vs. low volatility) on the filtering performance?

---

> ### Author Response · Authors · 2024-11-20
> **Re: Reviewer bkmt [1/2] -  Choice of Datasets**
>
> Dear Reviewer,
>
> We deeply appreciate your thoughtful feedback and strong endorsement of our work, particularly your recognition of its **originality** and **creativity** in fusing classical control theory with modern machine learning. This positive assessment strengthens our commitment to ensure our work permeates and benefits the wider AI/ML research community and beyond.
>
> ### Responses to Weaknesses:
>
> > **[W1]** The financial market experiments focus primarily on a single dataset (NIFTY), and broader validation across different types of time series problems would strengthen the claims of **generality**.
>
> We thank you for pointing this out. As highlighted in our submission, we deliberately designed our experiments to span two fundamentally different time-series forecasting (TSF) tasks: the financial market movement (**FMM**) task and the electric grid long-term load forecasting (**LTSF**) task. These tasks were chosen for their orthogonality in dimensions such as domain, features, observation windows, forecasting horizons, stationarity, and evaluation metrics, illustrating the broad applicability of our approach.
>
> If we understand your concern correctly, it pertains to the usage (or lack thereof) of additional datasets **within the FMM task**.  While the financial domain indeed offers diverse datasets (e.g., commodities, cryptocurrencies, economic indicators), we deliberately opted to focus on demonstrating the orthogonal use case of LTSF. This decision was driven by two factors:
>
> 1. _Illustration of two loss paradigms_: Adding LTSF allowed us to showcase our method's efficacy across both `cross-entropy loss` (classification in FMM) and `MSE`, mean squared error (regression in LTSF) cases.
>
> 2. _Avoiding financial domain bias_:  Incorporating more financial datasets could overshadow our primary contribution - the relative performance guarantees and flexibility of the MoE-F framework - by the overt focus on downstream financial domain-specific results that are usually means to an end.
>
> Nevertheless, we acknowledge that additional FMM datasets could further validate our claims within this domain. We plan to incorporate these datasets in future work and appreciate your suggestion.

---

> ### Author Response · Authors · 2024-11-20
> **Re: Reviewer bkmt [2/2] -  Scalability Strategies, Market Regimes**
>
> ### Responses  to Questions
>
> > **[Q1]**  How does the performance of MoE-F degrade with an increasing number of experts, and what are the computational scaling implications?
>
> The theoretical runtime of our filtering algorithm is $O(N^2)$, but in practice, the number of experts is typically limited (~7 in our work). Thus, the (algorithmic) scaling implications are negligible for realistic use cases.
>
> Scaling concerns primarily stem from the computational cost of expert inference, especially with large-scale, billion-parameter LLMs.  To elucidate, simultaneously hosting a stream of 7 billion-parameter LLM experts requires significant inference compute infrastructure. However, MoE-F itself introduces no significant overhead, as it serves as a lightweight filtering harness applicable to a heterogeneous mix of experts.
>
> To address potential scalability concerns:
>
> - A top-k experts strategy can maintain a fixed computational budget, with less effective experts relegated and replaced periodically by offline or new experts.
>
> - Cost-effective classical models (e.g., ARIMA) can complement large-scale experts in low-resource settings.
> This modularity ensures that MoE-F adapts to a wide range of computational constraints.
>
> > **[Q2]** What is the impact of different market regimes (high volatility vs. low volatility) on the filtering performance?
>
> Market volatility estimation, while challenging, does not directly impact our method. Unlike traditional filtering setup where the measurement process reflects stock prices and volatility is the unobservable, latent signal process affecting the stock price. This does not apply here because we consider a single non-random path (similarly to the rough path integration, Rough Differential Equation (RDE) literature). MoE-F evaluates the performance of experts, with the "hidden signal" being the identity of the best expert at any time.
>
> Nevertheless, if regime changes occur rapidly (e.g., small occupation time of the hidden Markov process, i.e. frequent switching of the best expert), any statistical method, including MoE-F, may struggle. Conversely, in stable regimes where the expert performance ratios remain consistent, MoE-F is well-suited to approximate the best expert. Our mechanism relies on the stability of these performance patterns to function effectively, which aligns well with situations where market dynamics are not too erratic in nature.
>
> ---
> Finally, we would like to respectfully request that you reconsider the ranking of our paper. Given its originality and the strength of its theoretical and practical contributions, we believe it could be a valuable addition to the oral or highlight sessions at the conference. Presenting in such a format would enable us to reach a wider AI/ML audience, sparking broader discussions about the fusion of classical control theory with modern machine learning paradigms. We would be deeply grateful if you could consider this when finalizing your assessment.
>
> Thank you once again for your valuable feedback and consideration.

---

> > ### Comment · Reviewer_bkmt · 2024-11-25
> >
> > Thank you for your thorough response. Your explanation has addressed most of my concerns. I will discuss with other reviewers about potentially revising the score upward.

---

### Official Review · Reviewer_5hye · 2024-11-02

**Soundness:** 1
**Presentation:** 1
**Contribution:** 3
**Rating:** 3
**Confidence:** 3

**Summary:**

This work proposes to combine the predictions of multiple (large language) models in order to predict univariate time series. To optimise the weightings associated with each model based on past performances, the authors employ a continuous-time finite-state hidden Markov model (and an associated filtering algorithm).

**Strengths:**

**Novelty:** To my knowledge, this methodology is novel.

**Significance/contribution.** Making use of multiple (large language) models and using a filtering algorithm to decide in an online manner how to weight the predictions from each of these seems like a sensible idea.

**Weaknesses:**

**Major comments:**

1. **Clarity needs to be drastically improved.** I believe Sections 1, 2 and 3 need to be substantially rewritten to improve clarity. I do not believe that this manuscript is ready for publication in ICLR based on the presentation alone. For instance:

    - There seems to be no clear structure/order to the explanation of the model/methodology in Sections 1--3.
    - Key aspects of the methodology (including some notation) are explained in a figure caption (of Figure 2) and this figure is then not even referenced until the following page.
    - Important parts of the model, e.g., the processes $W$, $w$ or $x$ are used for several pages, before being defined in the middle of a generic-seeming paragraph titled "Probability Theory" at the end of Section 2.
    - Section 2, in particular, is a collection of disjointed paragraphs lacking a clear structure.
    - Sections 1 and 2 switch between continuous time and discrete time without explanation. Indeed, a time-discretisation is briefly mentioned in Section 3 but without sufficient detail.
    - Enumerating a bunch of "helper functions" (Equations 2--4) at the reader without much intuition or context makes for a difficult read.

    I think these sections should start with a high-level explanation of the (Wonham-Shiryaev) filtering algorithm and the model that this filtering algorithm is targetting. The specification of the model should be consolidated in one place.


2. **Undefined/poorly defined notation.** Related to the previous point, there is quite a bit of undefined or poorly defined notation (or notation that is only defined much later after it has already been extensively used). For instance:

    - In the caption of Figure 2 (and also in Line 9 of Algorithm 1), what are "$\pi_n$" and "$\pi^n$"? Is this the same as "$\pi_t^{(n)}$"?
    - In Line 178, what does "$Q_t^{n:i,j}$" mean?
    - In Line 5 of Algorithm 1, what is "$\ell$"? Page 2 defines a loss function $\ell$ but this has two arguments rather than one.
    - In Line 15 of Algorithm 1, what does "$P^{(n)}$" mean?
    - In Line 305, what does "$P_{w_{t+1}|w_{t-H:t}}$" mean?

3. **Soundness.** The presentational issues outlined above have made it impossible for me to verify that the proposed algorithm is valid in any meaningful sense. Hence I have to give a low score here.



**Minor comments:**

* Abstract: "remarkable" does not seem very objective.

* Figures should not appear in the text before they have been referenced. For instance, Figure 1 seems not to be referenced at all; Figure 2 is only referenced a page after it is displayed.

* The legends/annotations in some of the figures (e.g. Figure 1) are too small.

* Avoid "forward" referencing of equations, i.e. referencing equations before they appear in the text (see, e.g., Lines 81, 88, 191).

* Switching between bold and non-bold symbols for some of the mathematical quantities in Section 2 is confusing.

* The sentence "Each $n$th expert additionally provides their best ranking $\pi_t^{(n)}$ of the reliability of each expert’s performance thus far" in Lines 83--84 is very ambiguous (but seems central to understanding the proposed methodology).

* L284: The last sentence (starting with "Furthermore, ...") does not seem to contain a statement.


**Typos:**

* Abstract: Wohman-Shiryaev -> Wonham-Shiryaev

* L81: "predictions" -> "prediction"

* Eq. 2: Presumably, the time subscript $s$ (rather than $t$) is a typo?

* L163: The name "transition matrix" is confusing since this would normally be understood to mean $\mathcal{P}_N$. Presumably, "transition-rate matrix" or "intensity matrix" or "$Q$-matrix" is meant here.

* L165: Shouldn't this be "sum to 0" instead of "sum to 1"?

* L173: "Eq. equation"

* L220: missing space

**Questions:**

1. Regarding the loss functions discussed in Lines 129--134, can you explain how the classification scenario works? This would require $Y_t$ to be in $[0, 1]$. But I don't see how this can be guaranteed with the Brownian motion component in Equation 1.

2. Can you explain why we have "$w_t$" rather than "$w_s$" in Equation 1 and Line 101?

3. Why does the model need to be specified in continuous time (thus complicating the methodology). For the applications considered in this work, wouldn't a discrete time model, i.e. a simple discrete-time hidden Markov model, suffice?

---

> ### Author Response · Authors · 2024-11-20
> **Re: Reviewer 5hye [1/2] : Addressing Presentation Concerns**
>
> Dear Reviewer,
>
> We firstly thank you for acknowledging the strengths of our work for its novelty of methodology and significance of contribution. We would like to respectfully address the concerns you raised regarding the structure and presentation of our manuscript.
>
> While the overall review is unexpected/unusual, upon careful examination, it appears the concerns raised primarily pertain to the structure and presentation of the manuscript, rather than an evaluation of its contributions or the soundness of our methodology. In our revised version, we have made significant efforts to address these presentation issues and hope this allows for a **fair assessment** of the work's core methodologies.
>
> ## Responses  to Weaknesses
>
> > [Major Comments] Clarity, notations and soundness
>
> We recognize that expectations around the presentation style can be subjective, and while we understand your preference for a different structure, we would like to highlight that our manuscript follows a standard format commonly seen in AI/ML conference submissions. If it was indeed a case of highly unorthodox presentation for ICLR, the issue would be reflected, flagged across the reviews immediately.
>
> Nevertheless, with our aim to make our work accessible, we have taken your feedback seriously and meticulously addressed all the raised concerns to improve the clarity of our manuscript within the constraints of the conference format. Specifically:
>
> 1. **Clarity and Flow**: We have restructured the introductory sections 1 to 3 to create a more linear and cohesive flow. Notably, we moved Figure 2 to Section 3 and integrated its caption content into the main text, making it easier to follow. We also ensured that important definitions and steps are positioned close to their usage, thereby reducing the need for forward referencing and improving readability. For example, the main two steps are now contiguous within Section 3, adjacent/close to the main algorithm, allowing easier juxtaposition and referencing.
>
> 2. **Notation and Definitions**: In response to your concerns about undefined or poorly defined notation, we have added explicit explanations earlier in the manuscript. Section 2 has been revised to include a clear introduction to the necessary notation and concepts in continuous time, ensuring consistency throughout the paper. The helper functions are now introduced following adequate preliminaries (like conceptual overview, loss functions) in section 3.
>
> 3. **High-Level Explanation**: We have added a high-level overview at the beginning of Section 3 to provide context for the filtering algorithm and its purpose. Additionally, we included a remark in Section 2 pointing readers to Appendix D for more in-depth background on the filtering methodology, for those interested in further details.
>
> **Notations**
>
> - The differing versions of $\pi_n$ are minor typos meaning the same $\pi^{(n)}_t$ - now fixed.
> -  $Q^{(n:i,j})$ and $Q^{(n)}$ (line 174) are deprecated notation, it should read $Q^{i,j}_t$ as updated in our revised draft (line 125).
> -  You are correct, this is a typo:  $Y_t$ has to be given to $\ell$ as second argument.
>  - With $P^{(n)}$ we meant the n-th row of the matrix $P$. We changed to the more standard notation $P_{(n,\cdot)}$.
> -  $P_{w_{t+1} | w_{t-1+H}}$ should read '_on the time window $[t-H+1,t]$_'
>
> **Soundness**
> As you have acknowledged, you were unable to do a fair assessment of the soundness based on raised presentation issues. We hope our revised draft resolves this impediment, and we very much look forward to your updated thoughts.
>
>
> > [Minor comments and Typos]
>
> While we respectfully do not agree with every minor comment raised, we have nonetheless addressed and fixed all these (minor comments, typos) concerns. We are happy to address any further cascading/remaining concerns.

---

> > ### Author Response · Authors · 2024-11-20
> > **Re: Reviewer 5hye [2/2] : Questions**
> >
> > ## Responses  to Questions
> >
> > > [Q1] Regarding the loss functions discussed ... component in Equation 1.
> >
> > Though the setting in Equation 1 directly applies to unbounded targets, like in regression, one could also use it in classification by simply taking $Y_{\cdot}$ to be the logits (inverse logistic transform) of the classification probabilities.  Since the logistic function is a bi-measurable bijection, the filtration generated by the logits or the classification probability processes are equal; thus, the filter does not change. Further, we have included this explanation now under 'Loss Functions' [line 161-164] in S2 Preliminaries.
> >
> > > [Q2] Can you explain why we have "wt" rather than "ws" in Equation 1 and Line 101?
> >
> > This was a typo that we fixed now, thanks.
> >
> > > [Q3] Why does the model need to be specified in continuous time ... suffice?
> >
> > All discrete-time problems can be embedded into continuous-time ones via simple processes.  Thus, we consider continuous-time since it is **strictly more general**. It is quite common to do this in fields like stochastic calculus, since the continuous-time perspective often offers an easier view on the problem.
> >
> > We hope these changes address your concerns and allow for a fair reassessment of the manuscript's core contributions. We are confident that our work presents an important and novel advancement, and we are committed to making it as accessible as possible within the space constraints.
> > Thank you again for your feedback, and we look forward to your updated thoughts.

---

> ### Comment · Reviewer_5hye · 2024-11-22
>
> Thank you for your replies.
>
> I have had a look at the revision and I will increase my score since some of my comments (especially the typos, although I am still finding some (other) typos) have been addressed.
>
> However, some of the main points in my "Clarity needs to be drastically improved" comments have not been satisfactorily addressed. See especially the last recommendation about starting from a high-level explanation of the idea behind the algorithm. At the moment, the paper simply states an algorithm without sufficient intuition for why this algorithm would be valid or how it was derived. I don't think this is good enough.

---

> ### Author Response · Authors · 2024-11-22
> **Motivation and Intuition Behind MoE-F Algorithm Is Now Provided**
>
> Dear Reviewer 5hye,
>
> We appreciate your constructive feedback and agree that more motivation and high-level intuition for the MoE-F algorithm would enhance clarity.  We have revised the introduction to increase the transparency of our  MoE-F algorithm and its motivation.  Additionally, we have moved the formal definitions of the signal, measurement process, and loss to Section 3, as these elements formalize our setup but do not sufficiently convey the natural intuition behind it, which we aim to address earlier in the paper.
>
> Specifically, the updates to clarify the motivation and intuition for the MoE-F algorithm can be found on `lines 56-83` and `145-150`, with all modifications highlighted in **blue**.
>
> ### Distinguishing Static vs. Dynamic/Temporal Problems (Updated `Lines 61-66`)
>
> Classical Mixtures of Experts (MoEs) are typically designed for non-temporal (static) prediction tasks, where a fixed gating mechanism is used to route inputs.  In temporal prediction tasks, however, there is continuous feedback on the experts' performance, which can be leveraged to dynamically update the gating mechanism since this information indicates which expert model may be relevant and which may not be.  This key distinction is crucial in understanding how MoE-F operates.
>
> ### Rationale for a Filtering Approach in Temporal Settings (Updated `Lines 68-74`)
>
> In the temporal setting, the user constantly estimates the mixture coefficients, with their objective being the minimization of a prediction loss based on each expert's measured performance.  The mixture coefficients represent a partially observed signal, inferred indirectly through the experts' observed performance.  Hence, predicting these coefficients aligns naturally with the **continuous-time finite state-space stochastic filtering problem**, as described in [1].
>
> ### How the MoE-F Algorithm Works
>
> The MoE-F algorithm leverages this inherent stochastic filtering structure in temporal prediction problems.  It operates in two stages: first, $N$ parallel stochastic filters predict the optimal mixture weights over time based on the historical performance of each expert model.  These predictions are then aggregated into a single set of mixture weights by solving a robust optimization problem similar to those used in **PAC-Bayes theory** [2].
>
>
> We really appreciate your feedback on our writing style and we genuinely do believe it has strongly improved the quality of our presentation.
>
> Much appreciation,
>
> The Authors
>
>
>
> ### References
>
> [1] Wonham, W. Murray. "Some applications of stochastic differential equations to optimal nonlinear filtering." Journal of the Society for Industrial and Applied Mathematics, Series A: Control 2.3 (1964): 347-369.
>
> [2] Alquier, Pierre. "User-friendly introduction to PAC-Bayes bounds." Foundations and Trends® in Machine Learning 17.2 (2024): 174-303.

---

> > ### Author Response · Authors · 2024-11-27
> > **Follow-up**
> >
> > Dear Reviewer 5hye,
> >
> > We are looking forward to your response on the intuition we have provided to increase the accessibility of our results. We belive that your concerns have been addressed and we are looking forward to your thoughts on the matter.

---

### Official Review · Reviewer_qJNc · 2024-11-03

**Soundness:** 3
**Presentation:** 2
**Contribution:** 3
**Rating:** 8
**Confidence:** 2

**Summary:**

The paper proposes a filtering algorithm to combine predictions from several experts in sequential decision making problems. At each time-step each of the experts provide their prediction and their reliability based on their local estimates. Then based on the historical losses of the expert the algorithm comes up with mixture weights to combine the experts at time t and the process continues. The paper provides optimality guarantees for their method under a hidden markov model process environment. Then they show two applications: (i) combining LLM experts for the task of market direction prediction based on evolving text market data (ii) combining time-series experts for forecasting. They show remarkable gains in both applications.

**Strengths:**

1. The problem is interesting. The dynamic filtering method is easily implementable.

2. Both the applications are well selected and real world. The method shows a remarkable 17% gain over individual experts for the financial market direction task. This is pretty significant for a noisy problem.

**Weaknesses:**

1. The writing can be improved in some sections, specifically the implications of theorem 1 and 2 need to be explained better.

2. In the TSF application it is unclear at what granularity of time-steps is the filtering applied. For instance if the horizon is 720, do the individual experts predict every 60 time-steps and then their 60 step look-ahead predictions are combined and then we move to the task of predicting the next 60 time-steps. Or do they predict one step at a time for 720 time-steps?

**Questions:**

Asked in the weakness section

---

> ### Author Response · Authors · 2024-11-20
> **Re: Reviewer qJNc - Writing Improvements and Detailed Clarification of TSF Granularities**
>
> Dear Reviewer,
>
> We are sincerely grateful for your helpful feedback and kind words of encouragement. We have used your feedback to further enhance our manuscript for clarity and exposition. Below, we outline the specific actions we have taken to address the weaknesses you identified:
>
> ### Response to Weaknesses:
>
> > [W1]  The writing can be improved in some sections, specifically the implications of theorem 1 and 2 need to be explained better.
>
> We appreciate the feedback and agree that some of the writing could be better harmonized.
> As you will see in our updated manuscript, we have improved the exposition by streamlining the introductory sections to improve the clarity and flow better. (specific changes detailed below).  These changes, along with our additional feedback-based writings refactorings, should significantly enhance the flow and clarity of our manuscript.
>
>
> > [W2] In the TSF application it is unclear at what granularity of time-steps is the filtering applied. For instance if the horizon is 720, do the individual experts predict every 60 time-steps and then their 60 step look-ahead predictions are combined and then we move to the task of predicting the next 60 time-steps. Or do they predict one step at a time for 720 time-steps?
>
> This is a great question -  thank you for pointing out this ambiguity. The two plausible scenarios you described are essentially what the granularity categories in LTSF literature are dichotomized into:
>
> - (i) Iterated multi-step (**IMS**): predicting one step at a time over the horizon, $H$: $\hat{x}_{t+1}$,
> and
>
> - ii) direct multi-step (**DMS**): predicts the entire future window, $H$ in a single operation: $\hat{x}_{t+1:t+H}​$.
>
> In our experiments, we adopt the Direct Multi-Step (DMS) strategy for filtering, where each expert predicts the entire horizon, $H$, in one step.  The look-back window (or, lag period), $L$, is set to 720 time-steps, and our method follows a channel-independent (CI) strategy. This approach aligns with state-of-the-art (SOTA) Long-Term Series Forecasting (LTSF) architectures. With this setup, the Mean Squared Error (MSE) loss is then calculated across all channels $C$, as outlined by the  equation in S5.2 [Ln. 455] of our manuscript.
>
> To ensure clarity for all readers, we have added **Appendix Section E.2** [Ln. 1713] to the revised manuscript, where we discuss these distinctions in detail, including the differences between IMS and DMS strategies and their implications for our experimental setup.
>
>
> ### Changes to the Manuscript:
> In our updated manuscript:
>
> - We have streamlined the presentation of  notations, main steps, and helper functions definitions w.r.t. the theorems and algorithms to improve exposition and flow. Specific changes include, moving Figure 2 later in section 3, and incorporating the main steps (1,2) descriptions to within the paragraps from caption. The helper functions are defined later in S3, following preliminaries (notations, loss functions), adjacent to the main algorithm's presentation for easier juxtaposition and reference.
>
> - We added Appendix Section E.2 to discuss the granularity of time-steps in our time-series forecasting (TSF) experiments. This section explains our use of DMS and the channel-independent approach, as well as the rationale behind these choices.
>
> We are confident that these additions and revisions address your concerns and make our methodology and contributions more transparent. If you have further questions or need more clarification, please let us know.
>
> Once again, we greatly appreciate your constructive feedback and hope that our improved manuscript meets your expectations. Thank you very much for your time and for highlighting points that have allowed us to improve our work.

---

### Official Review · Reviewer_8aRL · 2024-11-03

**Soundness:** 3
**Presentation:** 3
**Contribution:** 3
**Rating:** 6
**Confidence:** 2

**Summary:**

This paper presents an online algorithm, called MoE-F, to adapt a mixture of pretrained large language models (LLMs) for time-series prediction tasks. A filter is used for each LLM, and the learning algorithm will learn those filters from data. The authors then provide an analysis about their method to reveal some nice properties of the solution. Finally, an extensive experiment has been done to evaluate their proposed method, using many big pretrained models, e.g., Llama-2, Llama-3, Mixtral, DBRX, ... Two types of problems are used in their evaluation, classification of financial market movement, and time-series forcasting. The results suggest that the new method can perform significantly better than the baselines for the first classification task. For time-series forcasting, their method also exhibits competitive performance.

**Strengths:**

**Originality:**
The proposed method seems novel and work well with time-series forcasting in an online fashion.

**Quality:**
The performance of their proposed method seems good, compatitive with the recent baselines. Their method is supported by some theoretical analysis.

**Clarity:**
The writing is readable with soem efforts for new readers.

**Significance:**
Please refer to other reviewers for this aspect, as I am not familiar with the topic of this paper.

**Weaknesses:**

The experimental results pose a concern about their way of evaluation or measurement. This concern comes from the strange behavior of F1 score in Table 2 and Table 3. In some columns (e.g., most columns in table 3), the value of F1 is smaller than both Precision and Recall. This is really strange. In my understanding, F1 is a hamonic mean of Precision and Recall, and hence it can not be less than both Precision and Recall.

**Questions:**

Can the authors explain about the concern before? Does it only a writing mistake?

---

> ### Author Response · Authors · 2024-11-20
> **Re: Detailed Clarification of  the F1-score Concern**
>
> Dear Reviewer,
>
> We thank you for recognizing the originality and strength of our work. We are especially grateful for your astute observation regarding the F1-measure discrepancies related to precision and recall values in Table 2 and Table 3 of our experimental results for the financial market movement (FMM) prediction task.
>
> > [W1] This concern comes from the strange behavior of F1 score in Table 2 and Table 3. In some columns (e.g., most columns in Table 3), the value of F1 is smaller than both Precision and Recall. This is really strange. In my understanding, F1 is a harmonic mean of Precision and Recall, and hence it cannot be less than both Precision and Recall.
>
> For the FMM task (evaluated on the NIFTY test split), the evaluation falls under the category of multi-class classification with three labels: "fall," "neutral," and "rise," for which the support is imbalanced across the labels (support: 73, 143, 101, respectively). This class imbalance necessitates a careful approach to evaluation metrics, particularly in averaging methods for precision, recall, and F1-score.
>
> We use the standard [scikit-learn](https://scikit-learn.org/1.3/modules/model_evaluation.html)​​ library methods for evaluation.
> The seemingly unexpected behavior of the F1-score being lower than both precision and recall arises from our use of _weighted_ averaging—a behavior that has been noted and documented in several discussions and issues [1](https://github.com/scikit-learn/scikit-learn/issues/83), [2](https://stackoverflow.com/questions/8284456/f1-smaller-than-both-precision-and-recall-in-scikit-learn).
>
> Under the 'weighted' averaging scheme, metrics for each class are computed separately, then averaged based on the class support (i.e., the number of instances for each label). In imbalanced settings, this approach can produce a global average where F1 is lower than both precision and recall due to the influence of poor performance on minority classes, effectively weighting down F1-score.
>
> To illustrate this phenomenon and provide additional clarity, we have added **Appendix E.1** section [Ln. 1679] in the revised manuscript, which includes the **confusion matrices** corresponding to Table 2. These confusion matrices clearly show how the performance varies across different label classes, providing a better visual understanding of the model's behavior. The F1-score's sensitivity to poor performance in any class is evident, and this sensitivity explains the observed values being lower than both precision and recall in some cases.
>
> Our updated manuscript now include updated Table caption [Ln. 378] and this explanation [Ln. 1681-1688], aiming to assist other readers with this astute observation. We have also updated our supplementary materials repository [1] with a script (`generate_results.sh`) that allows for the complete reproduction of the experiment results (with both `weighted` and `micro`  averaging schemes), in alignment with our commitment to reproducibility as per our "Reproducibility Statement."
>
> We hope that our detailed explanation and improvements to the manuscript sufficiently address your concern. If so, we kindly request that you reconsider your overall evaluation, as we believe our work, if allowed to present in a highlighted session at the conference,  will significantly bolster our contributions to reach a wider AI/ML audience  and future works in this promising new direction of better harnessing the ever-growing list of LLMs.
>
> **Reference**:
>
> [1] https://anonymous.4open.science/r/moe-f
>
>
> Once again, thank you for your detailed feedback and for highlighting an important point that has allowed us to strengthen the clarity of our work.

---

> > ### Comment · Reviewer_8aRL · 2024-11-26
> > **About F1**
> >
> > Thank you very much for your detailed explanation that removed my concern about F1 in the tables.

---

> > > ### Author Response · Authors · 2024-11-26
> > > **Re. 8aRL: Clarification of F1 ambiguity**
> > >
> > > Dear Reviewer,
> > >
> > > Thank you for your kind acknowledgment of our explanation regarding the F1 scores in the tables. We greatly appreciate your thoughtful engagement and the opportunity to clarify this nuanced point (that we missed initially), as we recognize how such details are critical to ensuring the clarity and integrity of our work.
> > >
> > > If there are any further aspects of the paper you believe could benefit from additional clarification or improvement, we are more than willing to address them. Your feedback has been invaluable, and we hope that the resolution of this issue reflects positively on your overall assessment of our work.
> > >
> > > We deeply value your insights and look forward to hearing if there’s anything else we can address to enhance the paper further.
> > >
> > > Sincerely,
> > > The Authors

---

### Meta-Review · Area_Chair_iZGW · 2024-12-23

**Metareview:**

This paper presents a mixture of experts (MoE) framework where the experts are pre-trained LLMs and the considered applications come from the domain of temporal prediction. The paper also contains theoretical analysis and optimality guarantees.

__Strengths__
This work is found to be significant and novel. The motivation is also found to be convincing: the classical MoE framework is developed specifically for the temporal domain and hence employs time-adaptive stochastic filtering techniques to combine experts. Or, as reviewer bkmt said, “it is a creative fusion of classical control theory with modern ML”. Finally, the experiments consider real-world applications and demonstrate substantial gains.

__Weaknesses__
While experimental evaluation is generally found to be satisfactory, at the same time there are some obvious further explorations that the authors could have considered to render the paper more complete - see suggestions by reviewer bkmt.

Secondly, more than one reviewers comment on the quality of writing, with reviewer 5hye providing multiple suggestions for improvement. I find that, to a large extent, the rebuttal has addressed some of these comments.

While a few remaining issues with the paper’s presentation remain as weaknesses of this work, I find that they are not significant enough to push the paper below the acceptance threshold. Overall, I find that the rebuttal has addressed most of the critical concerns of the reviewers and I am recommending this paper for acceptance.

**Additional Comments On Reviewer Discussion:**

As mentioned above, the rebuttal discussion focused a lot on clarifications requested by the reviewers. In certain cases, e.g. the question of reviewer 8aRL regarding an “inconsistency” with the F1 score, it was found that there was simply a misunderstanding which has now been cleared up.

A lot of the discussions also resulted in the manuscript being revised to be clearer and more intuitive to read (including the order in which insights and key concepts are presented).

---

### Decision · Program_Chairs · 2025-01-22

Accept (Poster)